# Tunable afterglow for mechanical self-monitoring 3D printing structures

Rongjuan Huang[1,2,6], Yunfei He[1,6], Juan Wang[1], Jindou Zou[1], Hailan Wang[1], Haodong Sun[1], Yuxin Xiao[1], Dexin Zheng[3], Jiani Ma[3], Tao Yu ✉[1,2] & Wei Huang ✉[1,4,5]

Self-monitoring materials have promising applications in structural health monitoring. However, developing organic afterglow materials for self-monitoring is a highly intriguing yet challenging task. Herein, we design two organic molecules with a twisted donor-acceptor-acceptor' configuration and achieve dual-emissive afterglow with tunable lifetimes (86.1–287.7 ms) by doping into various matrices. Based on a photosensitive resin, a series of complex structures are prepared using 3D printing technology. They exhibit tunable afterglow lifetime and Young's Modulus by manipulating the photo-curing time and humidity level. With sufficient photocuring or in dry conditions, a long-lived bright green afterglow without apparent deformation under external loading is realized. We demonstrate that the mechanical properties of complex 3D printing structures can be well monitored by controlling the photocuring time and humidity, and quantitively manifested by afterglow lifetimes. This work casts opportunities for constructing flexible 3D printing devices that can achieve sensing and real-time mechanical detection.

Organic long-lived afterglow materials have attracted remarkable interest for various applications in sensing[1], data encryption[2,3], anti-counterfeiting[4,5], organic optoelectronic devices[6,7] and bioimaging[8], due to their long emission lifetime and high exciton utilization[9]. Due to the spin-forbidden nature of transitions between states with different multiplicities in organic molecules, the spin-orbital coupling (SOC) process is very weak leading to a low triplet utilization. Room temperature phosphorescence (RTP)[10] and thermally activated delayed fluorescence (TADF)[11] are two mainstream mechanisms that can theoretically achieve 100% exciton harvesting and long-lifetime emission. In recent decades, enormous efforts have been devoted to stabilizing triplet excitons, such as crystallization[12,13], halogen bonding interactions[14], polymerization[15,16] and host-guest doping[17–19],

intrinsically by promoting spin-forbidden intersystem crossing (ISC) process and suppressing non-radiative deactivation pathways[20,21]. Owing to the structural diversity and processible sample preparation, constructing host-guest doping systems is more appealing, particularly in large-scale production and flexible device fabrication[22,23]. Both amorphous polymer[24–26] and small molecule[27,28] based doping systems have been reported with long-lived organic afterglow. Despite significant achievement, afterglow materials are rarely employed to construct precisely prescribed 3D geometries.

Currently, in terms of application and market share, additive manufacturing (also known as 3D printing) has emerged as a state-of-the-art technology for constructing highly complex 3D geometrical structures in a customised manner[29–31]. 3D printing has shown

[1]Frontiers Science Center for Flexible Electronics (FSCFE) and Xi'an Institute of Flexible Electronics (IFE), Northwestern Polytechnical University, 127 West Youyi Road, Xi'an 710072, China. [2]Key Laboratory of Flexible Electronics of Zhejiang Province, Ningbo Institute of Northwestern Polytechnical University, 218 Qingyi Road, Ningbo 315103, China. [3]Key Laboratory of Applied Surface and Colloid Chemistry, Ministry of Education, School of Chemistry and Chemistry Engineering, Shaanxi Normal University, Xi'an 710119, China. [4]Key Laboratory of Flexible Electronics (KLOFE) & Institute of Advanced Materials (IAM), Nanjing Tech University (Nanjing Tech), 30 South Puzhu Road, Nanjing 211816, China. [5]State Key Laboratory of Organic Electronics and Information Displays & Jiangsu Key Laboratory for Biosensors, Institute of Advanced Materials (IAM), Nanjing University of Posts and Telecommunications, 9 Wenyuan Road, Nanjing 210023, China. [6]These authors contributed equally: Rongjuan Huang, Yunfei He. ✉e-mail: iamtyu@nwpu.edu.cn; vc@nwpu.edu.cn

numerous unique advantages, such as design freedom without assembly, high applicability in various materials including polymers, metals and composites and environmental friendly in terms of materials and processes, which significantly promote its development in the applications of information hiding, soft robotics, sensors and flexible electronics[32–35]. The wide applications are mainly determined by the properties of the materials that are used for device construction. Polymer-based 3D printing structures have the merits of being light-weight, highly flexible, biocompatible and sensitive to external stimuli, endowing them with shape or property changes over time[36]. Recently, many 3D printing materials with excellent luminescent properties have been reported. For instance, several RTP-active organic salt compounds were developed by introducing heavy-atom-participated anion-$\pi^+$ interactions to construct 3D-printed white light lampshades[37]. 3D printable fluorescent resins with tunable optical properties were manufactured using a photopolymer-based 3D printing strategy[38]. Despite the advances, few materials have been successfully used to fabricate complex 3D devices with desired geometries to realize self-monitoring mechanical properties.

Self-monitoring is the inherent ability of a material to detect the occurrence of stress, deformation or damage without external sensors[39–41]. External sensors, such as strain gauges, fibre optics and accelerometers, generally implement traditional structural health monitoring (SHM)[42–44]. Their complex structures and high costs make materials with self-monitoring capabilities highly competitive. Currently, the most commonly used self-monitoring materials are piezoresistive composites, which are usually achieved by adding conductive materials to polymer matrices, i.e., carbon fibres[45,46], graphene oxide[47] and reduced graphene oxide[48] and carbon nanotubes[49]. The deformation of composites may lead to changes in the filler and conductive network and further increase the resistivity[50]. However, it still requires an external device as an auxiliary to detect the change, and the failure location is also difficult to accurately locate. With the development of afterglow chromophores, their low cost, high efficiency, contactless detection and high sensitivity make them ideal candidates as self-monitoring materials in many promising areas, especially 3D-printable complex devices, which may further enhance their competitiveness in the SHM field. Therefore, developing tunable afterglows to achieve real-time monitoring of specific properties of 3D structures is an attractive research topic.

In this work, we design and investigate two emitters, DTPPAO and tBuDTPPAO, with a donor-acceptor-acceptor' (D-A-A') configuration, where strong electron-donating diphenylamine (DPA) derivatives and electron-withdrawing diphenylphosphine oxide (DPO) are linked by a weaker acceptor unit dibenzothiophene (DBT). Several small molecules and polymers are chosen as host matrices to construct host-guest doping systems, achieving tunable TADF-RTP dual-emission. With decreasing triplet energies of hosts, RTP emission becomes dominant. An increased lifetime from 86.1 to 287.7 ms and 199.1 to 298.4 ms in DTPPAO and tBuDTPPAO doping systems, respectively, is observed. The triplet state of the host matrices shows a synergistic effect on the excited state of the guest molecules, which greatly enhances the ISC and energy transfer processes. Furthermore, by employing Digital Light Processing (DLP) 3D printing technology, a series of complex 3D structures are prepared by doping DTPPAO into an acrylic derivative, hydroxyethyl acrylate (HEA)-acrylic acid (AA) photosensitive resin. It is demonstrated that both ultraviolet (UV) irradiation curing time and humidity can quantitively control the mechanical properties by either manipulating the cross-linking degree of HEA-AA photosensitive resin or intermolecular hydrogen bonding interactions, which can be manifested by afterglow lifetimes. With sufficient photocuring time of 7200 s or in a dry environment, 3D printing structures exhibit bright green afterglow with a long lifetime lasting about 5 s and no apparent deformation with additional loading. This innovative and universal strategy opens a research path to manipulating molecular excited-state properties. It reveals the potential applications in real-time mechanical detection, sensing and flexible 3D printing devices.

## Results

### Molecular design and photophysical properties

Two molecules DTPPAO and tBuDTPPAO with D-A-A' structures studied in this work are presented in Fig. 1a. The rational design is based on the following considerations: (i) DPA is a commonly used strong electron-donating unit with a distorted structure; (ii) DBT with shallow the lowest unoccupied molecular orbital (LUMO) and deep highest occupied molecular orbital (HOMO) energy levels works as a weaker acceptor; the planar structure can increase molecular conjugation for a high photoluminescence (PL) efficiency. It has also been reported showing strong triplet formation properties;[51,52] (iii) DPO can form strong hydrogen bonding interactions between the P = O group and H atoms, and provide a steric hindrance reducing the possibility of triplet-triplet annihilation[53]. The reasonable selection and modification of D and A units is conducive to intra- and intermolecular charge transfer synergic dipole-dipole interaction and triplet formation. This design gives both molecules a twisted U-shape configuration, which can decrease the spatial orbital overlap and reduce aggregation-induced quenching for higher quantum yields. Frontier molecular orbital distributions present a clear orbital separation and twisted structures of DTPPAO and tBuDTPPAO (Fig. 1b). The HOMOs of DTPPAO and tBuDTPPAO are located on the D unit with a slight extension to the adjacent phenyl ring of DBT unit, while LUMOs mainly localize on the weak acceptor DBT unit, and LUMOs+1 mainly occupy the relatively strong donor DPO. This is due to the degeneracy of energy level[54,55], which provides an additional electron transfer channel from the DPA to the DPO unit. Therefore, both acceptor units (DBT and DPO) can contribute to the charge transfer process with this molecular configuration. Moreover, the weak intermolecular interaction between the D and A units (such as C-H⋯O, C-H⋯N and C-H⋯π) is not only beneficial to improve the order degree of the molecule in the amorphous state but also can increase the dipole moment of the molecule through electrostatic effect. The detailed synthetic processes of all studied compounds are presented in Supplementary Figs. 1 and 2. The structural characterizations and thermal analyses are shown in Supplementary Figs. 3–27 and Supplementary Table 1.

The low-lying energy levels and photophysical properties of compounds DTPPAO and tBuDTPPAO were first characterized in solutions. In diluted solutions, both molecules exhibit a weak and broad absorption band in the 350–400 nm range, which shows a bathochromic shift as solvent polarity increases (Supplementary Figs. 28, 29). It is attributed to the intramolecular CT transition, revealing a strong electronic coupling between D and A units in their ground state. Obvious redshifted PL spectra are also observed with increasing solvent polarity, further confirming the CT nature of the $S_1$ state. Compared with DTPPAO, tBuDTPPAO exhibits a slightly redshifted PL spectrum due to the enhanced CT by the tert-butyl substituents. Calculated from the onsets of the PL and phosphorescence spectra, the singlet-triplet energy gaps ($\Delta E_{ST}$) are 0.24 and 0.23 eV in DTPPAO and tBuDTPPAO (Supplementary Fig. 30), respectively, which are favourable for TADF emission.

The host-guest doping design strategy is then employed to construct organic afterglow systems. Here, triphenylphosphine oxide (TPO), triphenylphosphine (TPP) and diphenyl-sulfone (SF) are selected as the host matrices that can provide a rigid environment to increase intermolecular interactions and suppress triplet quenching (Supplementary Fig. 31). The DSC (differential scanning calorimeter) and TGA (thermogravimetry analyses) studies indicate that the melting points of the guest/host systems varied with the host materials (Supplementary Figs. 26, 27). Their low melting points provide the possibility of sample preparation via a conventional melt-casting method

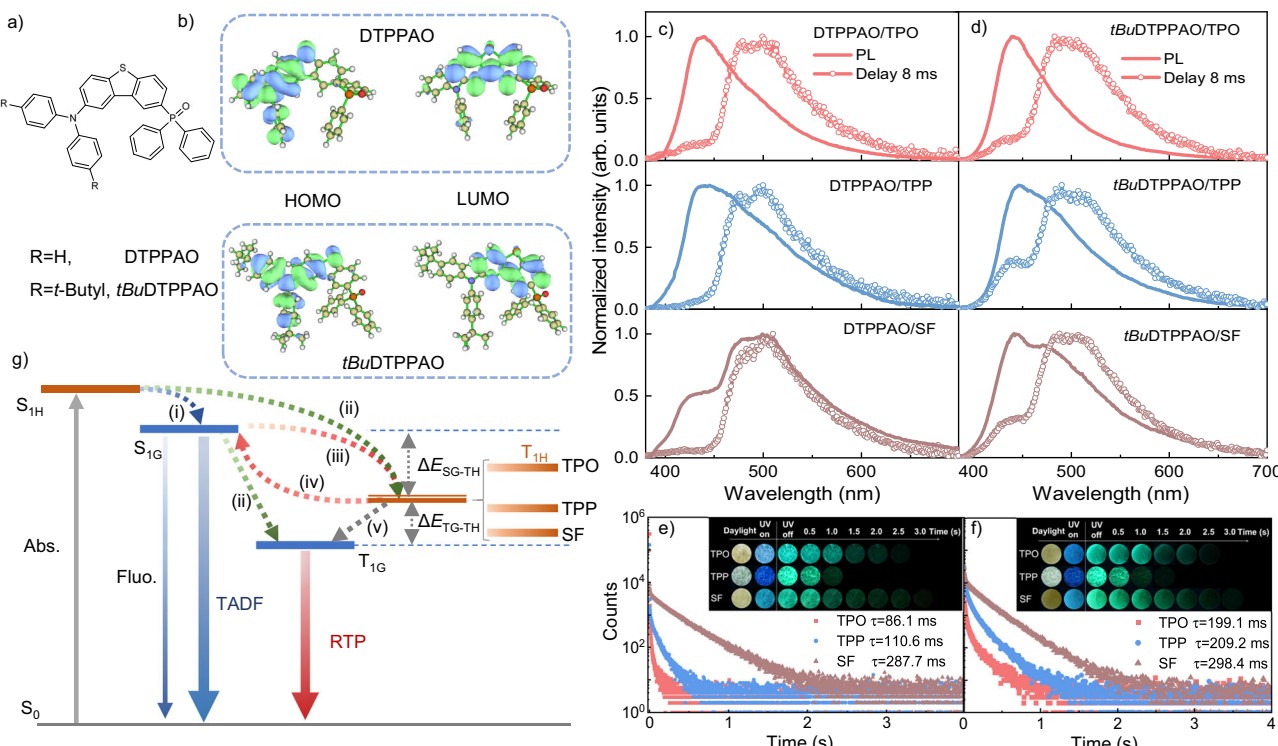

**Fig. 1 | Molecular structures and photophysical properties of DTPPAO and tBuDTPPAO doping systems. a** Chemical structures of the investigated molecules. **b** HOMO and LUMO distributions of DTPPAO and *tBu*DTPPAO calculated at the B3LYP/6-311 G(d) level. Normalized steady-state photoluminescence (PL) emission and delayed spectra (delay time of 8 ms) of (**c**) DTPPAO and (**d**) *tBu*DTPPAO in different host matrices at ambient conditions. Time-resolved decay curves of (**e**) DTPPAO/host systems at the phosphorescence emission band of 503 nm and (**f**) *tBu*DTPPAO/host systems at the phosphorescence band of 487 nm. Insets: Photographs of guest/host systems taken under daylight, 365 nm UV lamp on and off. **g** The schematic of mechanism and proposed transfer paths between guest and host molecules in the doping systems.

(Supplementary Table 2)[27]. As shown in Fig. 1c, the DTPPAO/TPO system exhibits a broad PL spectrum with an extended tail until 650 nm. At a delay time of 8 ms, a dual emission spectrum peaking at 442 and 495 nm is observed, which can be assigned as TADF and RTP emission, respectively, confirmed by the temperature-dependent measurements (Supplementary Fig. 32). The isoenergetic point also reveals the two bands of different emitting species. The millisecond range of the fluorescence band further confirms its long-lived TADF mechanism (Supplementary Fig. 33). In addition, the RTP band shows a well-resolved structure fitting well with the phosphorescence spectrum of DTPPAO, revealing its origin from the $T_1$ state of the guest molecule (Supplementary Fig. 34).

In the DTPPAO/TPO system, phosphorescence contribution in the PL spectrum is very weak. This is because the triplet state of TPO (3.07 eV) is closer to the singlet state of DTPPAO (3.05 eV), which provides the triplet excitons a more competitive rISC channel to $S_{1G}$ for TADF, while only less excitons can undergo energy transfer or vibronic coupling to the triplet state of DTPPAO ($T_{1G}$) for RTP. When doped into a host with a lower triplet state ($T_1 = 2.90$ eV of TPP and 2.83 eV of SF), an increased phosphorescence contribution is observed (Fig. 1c and Supplementary Fig. 35). It is most strikingly evident in the DTPPAO/SF system, which shows a large RTP/TADF ratio of 2.0, indicating DTPPAO/SF system as an RTP-dominated dual-emission. This is mainly due to the smaller energy gap between the triplet states of the host and guest molecules ($\Delta E_{TH-TG}$) in the DTPPAO/SF system (0.05 eV) from 0.13 eV in DTPPAO/TPO and 0.30 eV in DTPPAO/TPP system, which significantly facilitates the triplet-triplet energy transfer process for a strong RTP emission. The nonradiative deactivation pathways that strongly depend on the triplet energy gap ($\Delta E_{TH-TG}$) are also suppressed. Accordingly, the energy barrier between the singlet state of the guest and the triplet state of the host matrix ($\Delta E_{SG-TH}$) increases,

leading to a weaker TADF emission. With decreasing $\Delta E_{TH-TG}$, the lifetime of the doping system increases gradually from 86.1 ms in DTPPAO/TPO, 110.6 ms in DTPPAO/TPP to 287.7 ms in DTPPAO/SF as shown in the time-resolved decay curves (Fig. 1e). All DTPPAO/host systems exhibit bright green afterglow lasting for 1.5 ~ 3.0 s. Similar phenomena are also observed in *tBu*DTPPAO doped systems as shown in Fig. 1d–f and Supplementary Figs. 36–38. Compared with DTPPAO, all three *tBu*DTPPAO doped systems exhibit stronger TADF emission, mainly ascribing to the stronger CT character in *tBu*DTPPAO facilitating the rISC process. The long-lived afterglow in the guest/host system gives further evidence showing the rationality of our molecular design strategy, as its analogue D-A (DTPA) without an additional DPO acceptor unit shows much shorter RTP lifetimes in host matrices, which are 4.0 ms in DTPA/TPP and 149.8 ms in DTPA/SF (Supplementary Figs. 39–41). Moreover, compared with the doping systems in TPO and SF, a shorter duration time in the DTPPAO/TPP system is observed, probably due to the relatively higher PLQY (5.60%). The corresponding parameters are summarized in Table 1.

Based on the tunable dual-emissive afterglow properties in host molecules, we further explored the photophysical properties of DTPPAO in polymer matrices. By employing a commonly used polymer PMMA, transparent DTPPAO/PMMA and *tBu*DTPPAO/PMMA films were prepared, and both exhibit long-lived afterglow (Supplementary Figs. 42–47). However, due to the large singlet-triplet energy gap, both films exhibit weak TADF emission. The phosphorescence spectra of guest molecules in small organic matrices show a slight redshift compared with that in PMMA film. One reason could be the influence of the host polarity[56]. In addition to the increased rigidity that suppresses the non-radiative pathways, the small molecule matrices (TPO, TPP and SF) can also increase the polarity of the doping systems, resulting in a spectral redshift. Moreover, sufficient intermolecular

**Table 1 | Photophysical properties of DTPPAO and *tBu*DTPPAO (1 wt%) in different hosts**

|  | Host | $\lambda_{PL}/\lambda_{PH}$ (nm)[a] | $S_1/T_1$ (eV)[b] | $\Delta E_{ST}$ (eV)[c] | $\Phi_{PL}$ (%)[d] | $\tau$ (ms)[e] |
|---|---|---|---|---|---|---|
| DTPPAO | TPO | 440/500 | 3.05/2.77 | 0.28 | 4.01 | 86.1 |
|  | TPP | 440/500 | 3.06/2.77 | 0.29 | 5.60 | 110.6 |
|  | SF | 441/501 | 3.07/2.76 | 0.31 | 5.07 | 287.7 |
| tBuDTPPAO | TPO | 441/490 | 3.04/2.78 | 0.26 | 9.15 | 199.1 |
|  | TPP | 446/490 | 3.03/2.79 | 0.24 | 12.62 | 209.2 |
|  | SF | 442/490 | 3.04/2.78 | 0.26 | 8.38 | 298.4 |

[a]The peaks of fluorescence (RT) and phosphorescence spectra (77 K).
[b]Calculated from the onsets of the fluorescence (RT) and phosphorescence spectra (77 K).
[c]Experimentally determined singlet–triplet energy splitting.
[d]PLQY measured in air at room temperature.
[e]Lifetimes of phosphorescence spectra at room temperature estimated from the transient decay curves.

interactions between guest and small molecule hosts can also induce the spectral redshift[57]. All the possible reasons account for the spectral shift of phosphorescence synergistically. The photophysical properties are presented in Supplementary Table 3.

The mechanism of these host/guest doping systems is illustrated in a simplified energy diagram as shown in Fig. 1g. After photo-excitation, the excitons in the singlet state of the host matrix ($S_{1H}$) can either decay to the lowest singlet excited state of the guest molecule ($S_{1G}$) through Förster energy transfer process (i) or to its triplet state ($T_{1H}$) via ISC process (ii). Then, the singlet excitons of the guest molecule undergo an efficient radiative decay emitting fluorescence, which can also be converted to the triplet states of the guest ($T_{1G}$) and host ($T_{1H}$) molecules through intramolecular ISC process (ii) and singlet-triplet energy transfer (iii), respectively. The triplet state of the host matrix between $S_{1G}$ and $T_{1G}$ works as an intermediate energy-tunable platform, promoting excitons decay via (iv) rISC process upconverting excitons to the $S_{1G}$ for TADF emission and (v) triplet-triplet Dexter energy transfer to the $T_{1G}$ state for long-lived RTP[17]. It illustrates that a dual-emission with concurrent TADF and RTP characters can be obtained by properly selecting the host matrix, whose triplet state plays a synergistic role in the energy transfer process for tunable TADF/RTP ratios and lifetimes.

To further verify the triplet-triplet energy transfer, the excited state dynamics of the DTPPAO/SF system were investigated using femtosecond transient absorption (fs-TA) spectroscopy[58–60]. All the samples were measured using solid-state films upon excitation at 350 nm in ambient conditions, which may induce complex and broad signals in their respective TA spectra. Both SF and DTPPAO exhibit a broad positive absorption band in a large wavelength range. Upon longer delay time, the band at around 637 nm in SF shows a clear intensity increase from 139.77 ps, which can be assigned to the excited-state absorption (ESA) of $T_1 \rightarrow T_n$ (Supplementary Fig. 48a). Two distinctive ESA bands peaking at 575 and 620 nm appeared at 756.43 ps are identified as the triplet ESA bands of DTPPAO as they grow continuously in intensity and can live up to 2464.30 ps (Supplementary Fig. 48b). As shown in Supplementary Fig. 49, DTPPAO/SF doping system exhibits a broad positive TA band, including contributions from both DTPPAO and SF molecules. A clear spectral shift from 642 nm to around 600 nm is observed, corresponding to the triplet ESA bands of SF and DTPPAO, respectively. With increasing delay time, the intensity of the 642 nm band from SF decreases gradually, while the broad band (567-616 nm) from DTPPAO increases in intensity which appears earlier (~509.93 ps) than the pure DTPPAO film and lives up to a few nanoseconds. The intensity changes and gradual structured TA spectra of DTPPAO/SF provide qualitative evidence for distinct triplet-triplet energy transfer between host and guest molecules. Dexter energy transfer involves electron exchange and typically occurs within a short-range (<10 Å)[61,62], the concentration-dependent measurements including PL spectra and time-resolved decay curves were performed. As shown in Supplementary Figs. 50–55, all the

doping systems of DTPPAO and *tBu*DTPPAO exhibit a trend that phosphorescence emission enhances (500 nm in DTPPAO/SF and 490 nm in *tBu*DTPPAO/SF) with increasing doping concentration, while the time-resolved decay decreases correspondingly. It indicates that energy transfer becomes more efficient at a higher doping concentration, further confirming the triplet-triplet energy transfer process.

## Theoretical calculations

The frontier molecular orbital (FMO) distributions and excitation characteristics of DTPPAO and *tBu*DTPPAO were investigated by density functional theoretical (DFT) and time-dependent DFT (TD-DFT) calculations based on optimized molecular structures[63,64]. The HOMOs and LUMOs of TPO, TPP and SF matrices are all localized on their whole molecular skeleton (Supplementary Fig. 56). Due to the energy level degeneracy[54,55], LUMO and LUMO + 1 orbitals are mainly localized on DBT and DPO units, respectively (Supplementary Figs. 57, 58). The HOMO-LUMO energy gap of *tBu*DTPPAO is 0.12 eV smaller than that of DTPPAO, indicating its stronger intramolecular charge transfer (CT). The theoretically calculated singlet and triplet energy levels are 3.24/2.92 eV in DTPPAO and 3.15/2.84 eV in *tBu*DTPPAO. The lower energy levels can be attributed to the increased electron-donating ability of the substituted *tert*-butyl groups. Natural transition orbital (NTO) analysis of the singlet and triplet excited states was recorded as shown in Fig. 2 and Supplementary Figs. 59, 60. The $S_0$-$S_1$ excitations of both DTPPAO and *tBu*DTPPAO exhibit similar FMO distributions as the ground states ($S_0$). Notably, the holes are mainly dispersed on the donor unit, accompanied by a minor contribution from the phenyl ring and S atom of the DBT unit. While the particles are mainly distributed on the whole DBT unit in both $S_1$ and $T_1$ geometries, demonstrating both states have predominant CT characters in DTPPAO and *tBu*DTP-PAO. In contrast, their $T_2$ and $T_3$ states exhibit clear localized excitation characters (LE), with holes and particles localizing on the donor units. In both molecules, the triplet excited states ($T_1$, $T_2$, and $T_3$) are close to the $S_1$ state, indicating the existence of multiple ISC channels (from $S_1$ to $T_n$). Along with the small energy gaps, a moderate oscillator strength (*f*) and large spin-orbit coupling matrix elements (SOCME, ξ) between $S_1$ to $T_n$ reveal evidence of an effective ISC/rISC process for a favourable TADF channel. In addition, DTPPAO exhibits a relatively larger ξ($S_0$, $T_1$) than *tBu*DTPPAO, indicating a strong phosphorescence of DTPPAO from $T_1$. Therefore, dual TADF/RTP emission can be achieved in both DTPPAO and *tBu*DTPPAO, which is consistent with the experimental results.

## 3D printing structures in self-monitoring mechanical detection

Taking advantage of the tunable afterglow properties of DTPPAO, we employed HEA-AA (hydroxyethyl acrylate-acrylic acid) photosensitive resin as the polymer matrix and fabricated a series of complex and flexible perforated cubes by doping DTPPAO into HEA-AA via DLP 3D printing technology. HEA-AA is chosen as it can enhance

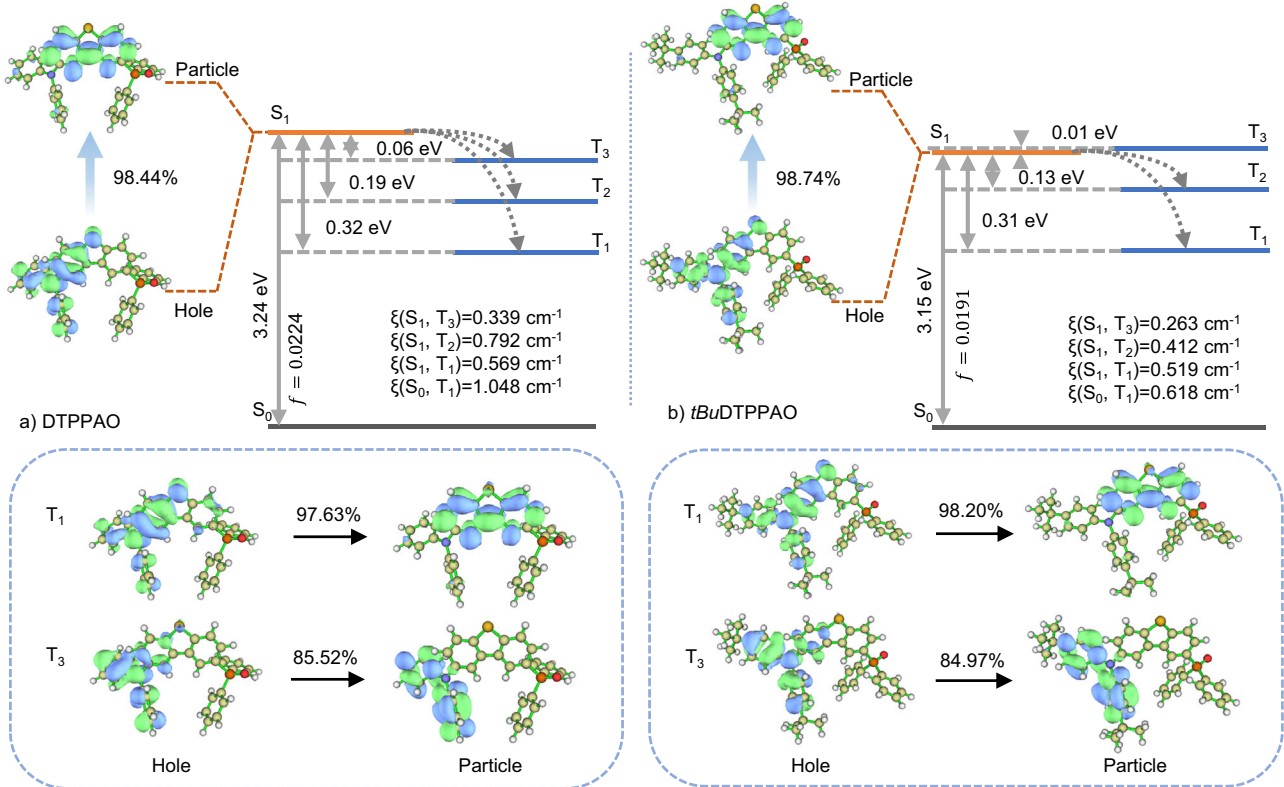

**Fig. 2 | Theoretical calculations.** Natural transition orbitals (NTOs) describing the excitation characters of the $S_1$, $T_1$ and $T_3$ states and vertical excitation energy levels with spin–orbit coupling matrix elements (ξ) calculated at the optimized molecular geometries in gas phase at the B3LYP/6-311 G(d) level of (**a**) DTPPAO and (**b**) tBuDTPPAO; the weights of the hole-electron contributions to the excitations are included.

intermolecular interactions and suppress nonradiative deactivations conducive to achieving long-lived afterglow. The luminescent and mechanical properties of DTPPAO/HEA-AA doping 3D printing perforated cubes exhibit high sensitivity to UV curing time and humidity.

To better understand the behaviours, we first prepared a series of DTPPAO/HEA-AA doped films (1 mm) and investigated the influence of photocuring time on the afterglow and mechanical properties. DTPPAO/HEA-AA film exhibits an obvious dual-emission (435/501 nm) after UV irradiation of 3 s (Fig. 3a). With increasing photocuring time, the intensity of RTP band increases gradually (Supplementary Fig. 61). Accordingly, RTP lifetime increases significantly from 1.7 to 526.5 ms (Fig. 3b, Supplementary Figs. 62–66). After sufficient photocuring (120 min), DTPPAO/HEA-AA film exhibits a bright green afterglow lasting more than 5 s. In contrast, the afterglow can be barely seen at a photocuring time of 10 s (Supplementary Fig. 67). Furthermore, the influence of polymer moisture content on the afterglow of DTPPAO/HEA-AA film by suspending it (photocuring time: 5 h) in a closed container which has a supersaturated NaBr salt solution (58%RH) at the bottom was investigated. The relative intensity of RTP in dual-emission shows a gradual decrease with increasing water-soaking time, and a decreasing lifetime from 489.1 to 2.9 ms accordingly (Fig. 3c, d, Supplementary Figs. 68–72). This is mainly because the moisture can break the hydrogen bonding interactions between DTPPAO and the HEA-AA matrix, which can be confirmed by the Fourier-transform infrared (FTIR) spectra as shown in Supplementary Fig. 73. The dry DTPPAO/HEA-AA film exhibits noticeable C = O and O-H bonds peaking at 1713 and 3456 cm$^{-1}$, respectively, which can be attributed to the associated hydroxyl group between adjacent DTPPAO and HEA-AA. After fuming with water vapour, the C = O and O-H peaks shift to 1706 and 3396 cm$^{-1}$, respectively, indicating an increase in the association degree of hydroxyl groups. Combined with the changes in afterglow properties, we demonstrate that the presence of water weakens and

breaks the hydrogen bonding interactions between adjacent DTPPAO and HEA-AA, resulting in significant activation of vibrational dissipation and consequent phosphorescence quenching[65,66]. In addition, the presence of dissolved oxygen in water can also quench triplet excitons, leading to decreased afterglow lifetime[67].

To understand the mechanism, the polymeric degree and the mechanical properties of DTPPAO/HEA-AA film were conducted. As shown in Fig. 4a, the perforated cubes exhibit no afterglow without or with a very short UV irradiation time, while an obvious bright green afterglow with increasing lifetime is obtained with increasing photocuring time. The conversion rate of acrylic acid C = C double bond of DTPPAO/HEA-AA film shows a continuous increase with increasing photocuring time, following the same trend as the afterglow lifetime (Supplementary Fig. 74). It indicates that the cross-linking degree of HEA-AA polymer chain can be enhanced upon the UV exposure on DTPPAO/HEA-AA film, which can significantly strength the rigidity of the doping system to suppress molecular vibrations and stabilize triplet excitons. Moreover, Young's Modulus also follows the same trend as the afterglow lifetime (Supplementary Figs. 75–77). As shown in Fig. 4b, the perforated cubes exhibit a short decay lifetime with almost no afterglow without UV irradiation curing and severe deformation under a weight of 2 N. With increasing photocuring time, bright green afterglow is observed, and the afterglow lifetime gradually increases ascribing to the enhanced polymer cross-linking degree. After sufficient polymerization (60 min photocuring), the perforated cube shows excellent mechanical properties without obvious deformation under a weight of 2 N, and the green afterglow lasts about 5 s (Supplementary Movie 1). It is demonstrated that the mechanical properties of complex 3D printed cubes can be manipulated by controlling the photocuring time, manifesting by tunable afterglow lifetime and intensity.

Furthermore, Young's Modulus of DTPPAO/HEA-AA film exhibits a noticeable decline from 890 to nearly 0 MPa with the increasing

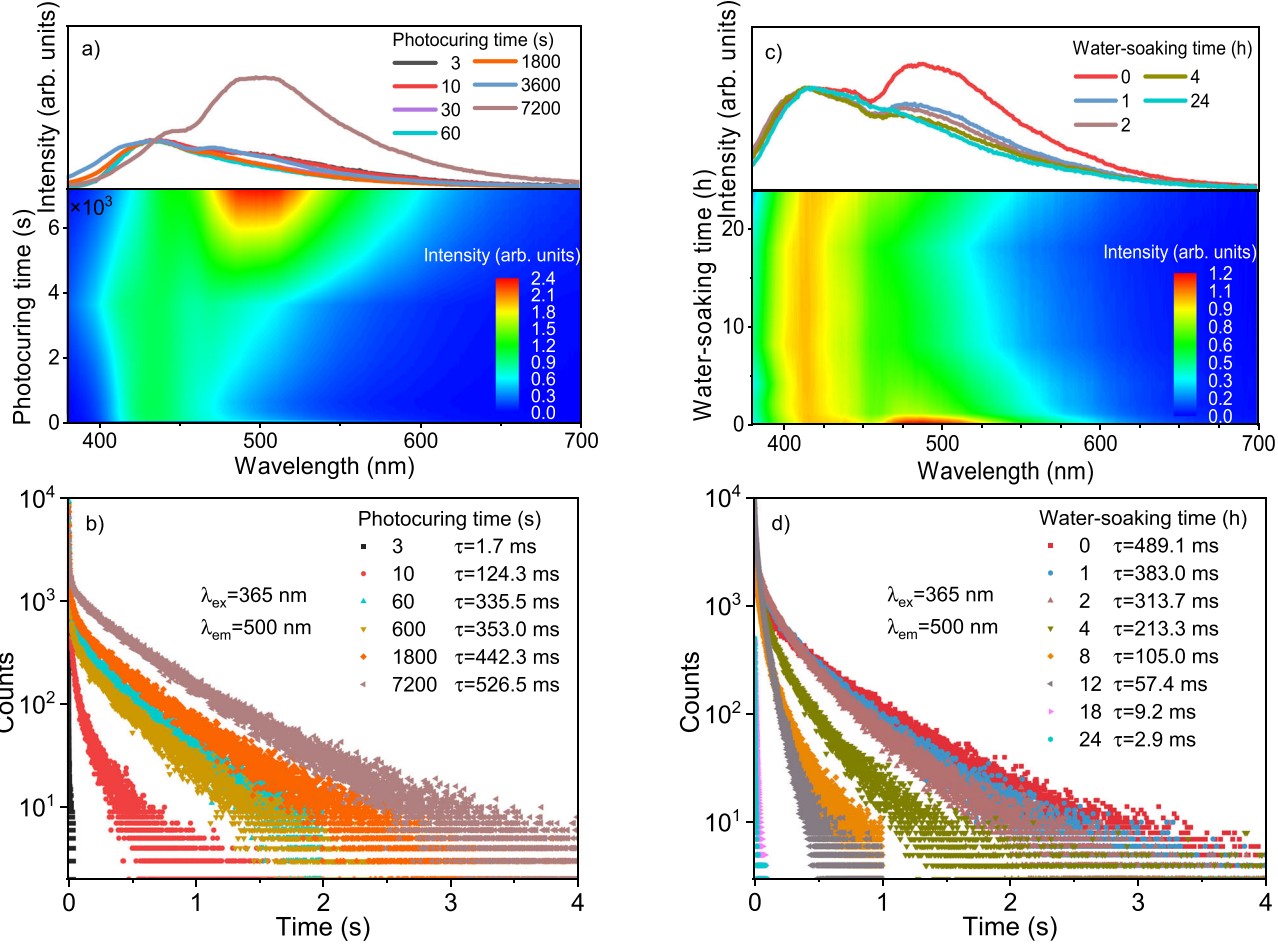

**Fig. 3 | Photophysical properties of 0.1 wt% DTPPAO/HEA-AA films (1 mm). a** Steady-state photoluminescence (PL) spectra and (**b**) lifetime decay curves of films at different photocuring time. **c** Steady-state PL spectra and (**d**) lifetime decay curves of films after 5 h photocuring treatment at different water-soaking time.

water-soaking time, which follows the same trend as afterglow lifetime (Fig. 5a, Supplementary Figs. 78–81). Compared to the mechanical properties and afterglow lifetimes with the same batch of perforated cubes after 5 h photocuring treatment, a severe deformation under a weight of 5 N in a moist environment was observed (Supplementary Movie 2). For better demonstration, we fabricated a table by DLP 3D printing, which was treated under 5 h UV irradiation before measurement. Then its two legs were put under 58%RH supersaturated NaBr solution as shown in Fig. 5b. As a result, only the two treated legs cannot hold the weight of 100 g and exhibit a short afterglow lifetime (Fig. 5c, Supplementary Movie 3). Moreover, the structure exhibits excellent fatigue resistance, as its phosphorescence lifetime shows a reversible change after five repeated cycles of water-absorbing and drying treatments (Supplementary Fig. 82). It further indicates that the hydrogen bonding interactions are weakened by the moisture in the doping system, resulting in a decreased Young's Modulus. It is suggested that the mechanical properties of DTTPAO/HEA-AA film can be quantitively manipulated by moisture levels and manifested by the afterglow lifetime. DTTPAO/HEA-AA doping 3D printing structures exhibit promising potential in extensive applications, including sensing, anticounterfeiting, data encryption and flexible 3D printing.

## Discussion

In summary, we designed and synthesized two twisted D-A-A' emitters, DTPPAO and *tBu*DTPPAO, with selectively distorted or planar electron-donating (TPA derivatives) and withdrawing units (DBT and TPO). Experimental results and theoretical calculations indicate that both molecules have a strong CT character. Tunable dual-mode TADF and

RTP emissions were obtained when doping into three selected small molecular host matrices, TPO, TPP and SF. With the decreasing triplet energy levels of the host matrix, the ratio of RTP/TADF intensity increases gradually from 0.5 in DTPPAO/TPO to 2.0 in DTPPAO/SF, correspondingly, their lifetime increases from 86.1 ms to 287.7 ms. Similarly, *tBu*DTPPAO doped systems also exhibit an increased RTP contribution and lifetime from 199.1 in TPO to 298.4 ms in SF. At the same time, a relatively strong TADF is observed in all three doping systems due to the *tert-butyl* substituents enhancing its CT character. This illustrates that the afterglow properties of DTPPAO and *tBu*DTP-PAO can be finely tuned by rationally doping into host matrices with proper triplet states, which can not only act as a rigid matrix suppressing the nonradiative deactivation but play an important role in the ISC and energy transfer processes.

Moreover, we further prepared a series of 3D printing structures based on DTPPAO by employing HEA-AA photosensitive resin as the host matrix. The dual-mode afterglow properties can be manipulated by controlling the photocuring time that influences the cross-linking degree of the HEA-AA and the polymer moisture content that induces abundant hydrogen bonds in the doping photocurable resins, which quantitively reflect the mechanical properties. The 3D structure exhibits no afterglow and severe deformation under pressure without UV photocuring, while a bright green afterglow is observed with high rigidity without deformation under a pressure of 2 N after a sufficient photocuring time of 60 min. Moreover, no deformation was observed even under a weight of 5 N in a dry condition. The afterglow lifetime and Young's modulus of DTPPAO/HEA-AA structure decrease gradually with increasing water-soaking time. It is demonstrated that the

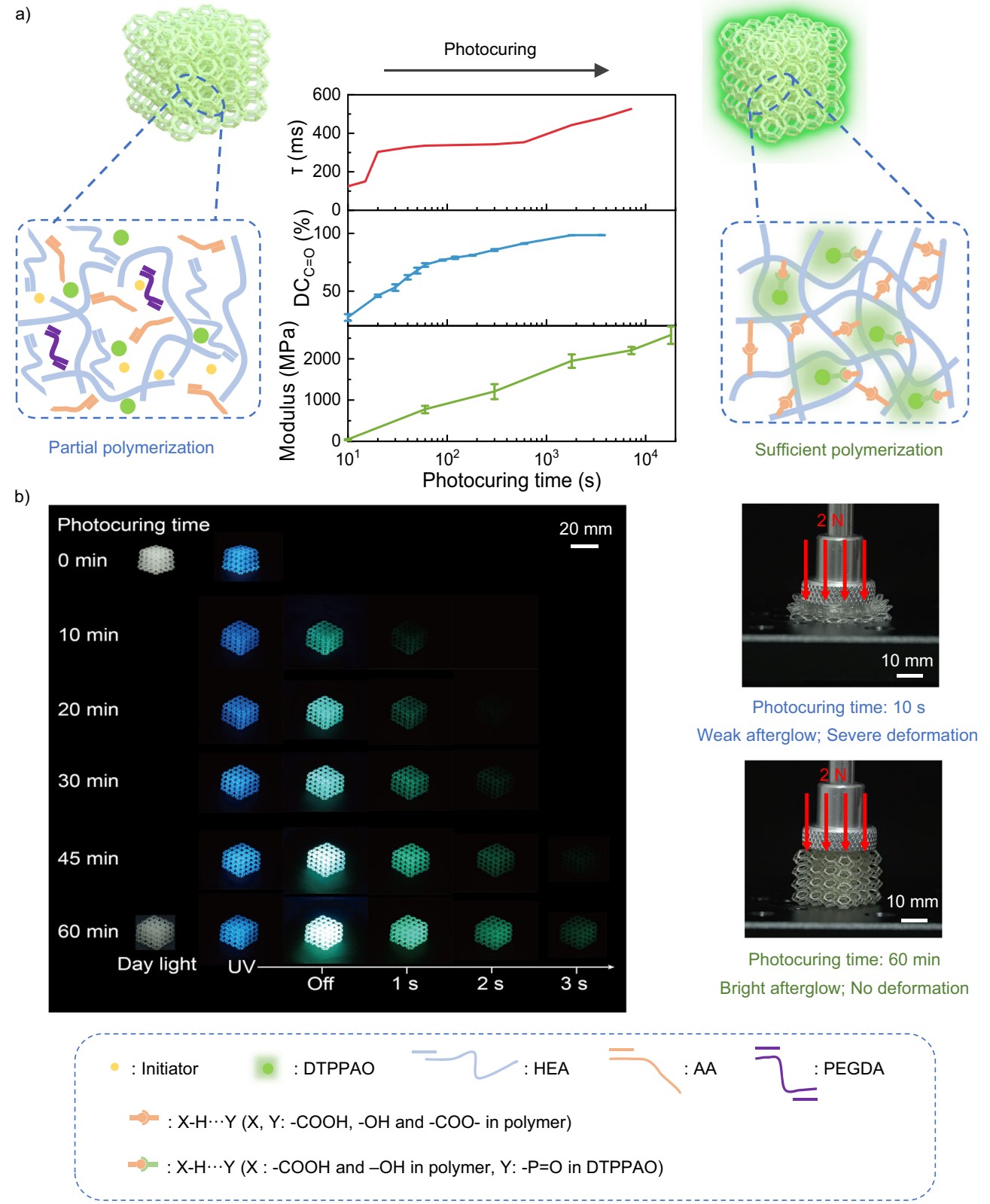

**Fig. 4 | Mechanical property monitoring upon photocuring time on 3D printed structures. a** C = C bond conversion rate (DC$_{C=O}$, upper), lifetime (τ, middle) and modulus (E, bottom) versus photocuring time upon 405 nm excitation, and the corresponding schematic polymer structure and perforated cubes afterglow. **b** Digital photographs of complex 3D printing perforated cubes at different photocuring times taken under a 365 nm UV lamp on and off (left) and the deformation under 2 N loading weights (right).

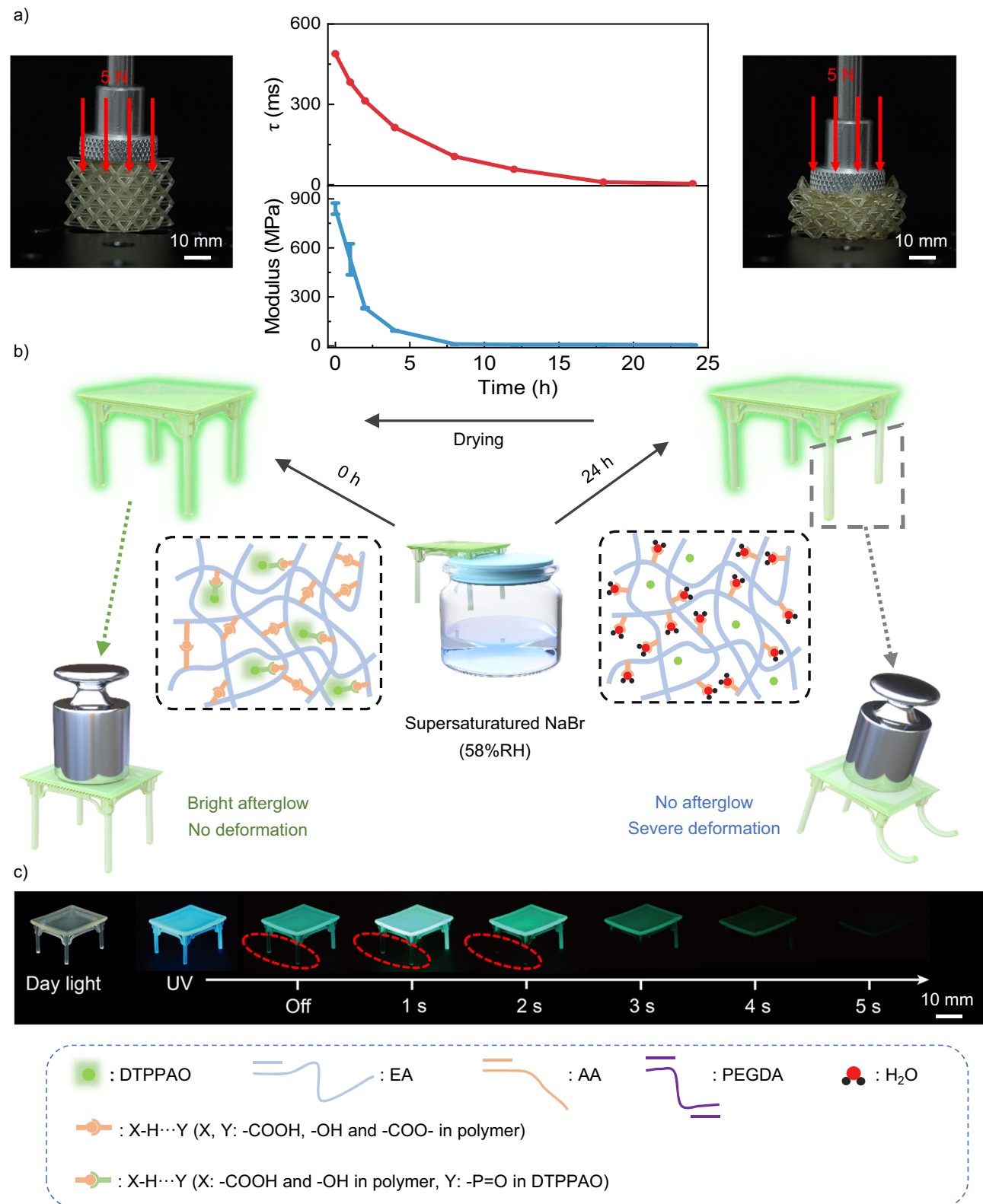

**Fig. 5 | Mechanical properties upon humidity of 3D printed structures.**
**a** Lifetime and Young's modulus versus water-soaking time and corresponding digital photos of 3D printed cubes (left: water-soaking time 0 h; right: water-soaking time 24 h) under the weight of 5 N loading. **b** Schematic diagram of local failure monitoring of Digital Light Processing (DLP) 3D printing table structure. **c** Digital afterglow photos of a partially failed table structure taken under a 365 nm UV lamp on and off.

mechanical properties of complex 3D printing structures can be quantitively manipulated by changing the photocuring time and moisture levels and manifested by the afterglow lifetime. This work shows the great potential of organic afterglow molecules in the application of flexible 3D printing and sensing for mechanical self-monitoring.

## Methods

### Syntheses and characterizations

The synthetic routes and molecular characterizations of DTPPAO and *tBu*DTPPAO, including nuclear magnetic resonance ($^1$H and $^{13}$C NMR), high-resolution mass spectra, high-performance liquid chromatography (HPLC) spectra and element analyses (EA), are summarized in detail in the Supplementary Information. NMR spectra were recorded at ambient temperature using Bruker AVANCE NEO 500 spectrometers, with working frequencies of 400 and 100 MHz for $^1$H and $^{13}$C, respectively. The high-resolution mass spectrometry of intermediates and target compounds was performed using Water Xevo G2-XS Tof (Waters) and LCMS-IT-TOF (Shimadzu), respectively. HPLC was collected using UltiMate 3000 (Thermo Fisher Scientific) with 90% methanol and 10% water (v/v). EA was recorded using Elementar UNICUBE with CHNS mode. X-ray diffraction (XRD) analysis was conducted using D8 Advance (Bruker). TGA was measured using TG 209 F3 (NETZSCH) at a constant nitrogen flow rate of 20.0 mL/min and heating rate of 20 °C/min from 30 to 900 °C. DSC measurement was performed using DSC214 (NETZSCH) at a constant nitrogen flow rate of 20.0 mL/min and heating rate of 10 °C/min from −40 to 150 °C.

### Theoretical calculations

All calculations were carried out using the Gaussian 09 program package. The ground state and the electronic geometries based on single-crystal structure were theoretically optimized using density functional theory (DFT) calculations at the B3LYP/6-311 G(d) level[64].

### Preparation of host/guest doping systems

Host materials were purchased from Damas-Beta (TPO, 98% and TPP, 99%) and Aladdin (SF, 97%). All the chemicals were purified by column chromatography and followed by recrystallization. Melt-casting: Host matrix molecule powder (1.0 g) and designed molecule (10.1 mg) were mixed in a round flask, and pumped nitrogen three times for degassing before heating up. The mixture was then slowly cooled down to room temperature after the blended powders were melted to a solid solution completely, eventually forming a solid-state doping sample. Drop-casting: PMMA powder (100.0 mg) and designed molecules (1.0 mg) were added into 5 mL toluene and stirred at 50 °C until blended powders were dissolved to a clear solution. The mixture was then drop-casted onto a quartz substrate and annealed at 80 °C for 2 min.

### HEA-AA doping films and 3D printing process

The photocurable resins were prepared by using hydroxyethyl acrylate (HEA) and acrylic acid (AA) as monomers and 1.5 wt% PEGDA as the crosslinker. 1.5 wt% TPO was added to the total amount of monomer and crosslinker as photoinitiator and 0.1 wt% DTPPAO was incorporated into the resin as a guest molecule with magnetic rotor stirring for 10 min to obtain a homogeneous solution of precursors. The quantitative solution was transferred to a Teflon mold with a pipette gun. The films with different curing times were obtained by curing the mold in a UV oven (BMF G2, China) for different time treatments. Samples with different moisture contents were obtained by placing the samples (photocuring time: 5 h) prepared by the above method in a suspension of supersaturated NaBr salt solution in an airtight environment to absorb water.

3D structures were performed on a commercial digital light process (DLP) 3D printer (nanoArch P150, BMF Material, China) with a 405 nm light source. The model was processed with 3Dmax software, and then the Stl-file of the model was sliced with BMF slice software to obtain 2D images (1920 × 1080) with a thickness of 50 μm. After printing, post-processing was performed by ultrasonically cleaning the printed structure with ethanol to remove residual resin and then placing the structure in a photocuring oven to cure for a certain period.

### Photophysical measurements

UV-vis absorption spectra were measured using a Shimadzu UV-3600 spectrophotometer. Steady-state emission spectra, delayed emission spectra and time-resolved decay lifetimes were recorded on an Edinburgh FLS1000 spectrometer equipped with a white light source and laser light sources (280 and 365 nm). Temperature-dependent measurements were acquired using a liquid nitrogen cryostat. Absolute photoluminescence quantum yields were also collected on an Edinburgh FLS1000 spectrometer equipped with an integrating sphere. Digital photos and videos were recorded by a Sony LICE-6400M camera. Fourier Transform Infrared Spectroscopy (FTIR) results were recorded using ATR mode with a Bruker Tensor II collecting 32 scans from 400 to 4000 cm$^{-1}$.

### Transient absorption spectroscopy

The fs-TA measurements were carried out based on a commercial regenerative amplified Ti:Sapphire laser system with an automated data acquisition system. A white continuum light (330–800 nm) was used as the probe pulse, which was generated in a CaF$_2$ crystal by about 5% of the amplified 800 nm output obtained from the laser system. The probe pulse was divided into two beams: one would pass the film sample, and the other was used as a reference to monitor the stability of the probe pulse. Here a 350 nm laser beam was employed to excite the sample.

### Mechanical measurements

The mechanical properties were studied in tensile mode using a Dynamic Mechanical Analysis (DMA) analyzer (Netzsch 242E, Germany). All samples had dimensions of 15.0 mm × 2.0 mm × 1.0 mm (length × width × thickness) and were measured at room temperature at a frequency of 1 Hz and an amplitude of 50 μm. Tensile testing was performed using CMT-8502 (500 N, Shenzhen Xinsansi Metrology) with samples of 20.0 mm × 2.0 mm × 0.9 mm. All samples were repeated more than three times.

### C=C bond conversion of photocuring process

The conversion rate is calculated as shown in Eq. (1):[68]

$$\text{Conversion}(\%) = 1 - \frac{\text{int}_x / \text{std}_x}{\text{int}_0 / \text{std}_0} \tag{1}$$

where $\text{int}_x$ and $\text{std}_x$ are the integrations of the C = C peak (1594–1649 cm$^{-1}$) and C = O peak (1649–1764 cm$^{-1}$) with UV exposure time of $x$ second, respectively. $\text{int}_0$ and $\text{std}_0$ correspond to the integration of C = C peak and C = O peak at the initial state, respectively.

## Data availability

All data supporting the plots and other findings of this study are available in the manuscript and Supplementary Information files, and available from the authors upon request. Source data are provided in this paper. Source data are provided with this paper.

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

## Acknowledgements

We gratefully acknowledge the financial support from the Fundamental Science Centre Funds (62288102, W.H.), the NSF of China (62275217, T.Y.), the Fundamental Research Funds for the Central Universities, China Postdoctoral Science Foundation (2021M692624, R.H.), Key Research and Development Program of Shaanxi Province (2020GXLH-Z-010, T.Y.), Natural Science Basic Research Program of Shaanxi Province (2022JQ-583, R.H.), Chongqing Science and Technology Fund (cstc2020jcyj-msxmX0931, T.Y.), Guangdong Basic and Applied Basic Reuter Foundation (2021A1515010633, T.Y.), Ningbo Natural Science Foundation (202003N4060, T.Y.), Key Project of Ningbo Natural Science Foundation (20221JCGY010492, R.H.), the National Aerospace Science Foundation of China (2020Z073053007, T.Y.) and Natural Science Basic Research Programme of Shaanxi (2024JC-JCQN-51, T.Y.).

## Author contributions

R.H. and Y.H. contributed equally to this work. R.H., Y.H., T.Y. and W.H. conceived the main idea and designed experiments. R.H. performed the theoretical calculations. Y.H. and J.W. synthesized and characterized the compounds. Y.H. and J.Z. performed the 3D printing structures preparation and mechanical experiments. R.H. and J.W. performed and analyzed the photophysical measurements. R.H., Y.H., D.Z. and J. M. performed and analyzed the transient absorption measurements. R.H. and Y.H. analyzed all data and wrote the manuscript with the help of H.W., H.S., Y.X., T.Y. and W.H. All authors discussed the results and commented on the manuscript.

## Competing interests

The authors declare no competing interests.
