## [Peer Review File · Nature Communications]

Tunable afterglow for mechanical self-monitoring 3D printing structuresREVIEWER COMMENTS

Reviewer #1 (Remarks to the Author):

In this manuscript, the authors report two novel D-A type emitters DTPPAO and tBuDTPPAO. When DTPPAO and tBuDTPPAO were doped into small molecules or polymer, tunable TADF-RTP can be achieved. The authors declared that the triplet states of the host matrices show a synergistic effect to the excited states of the guest molecules. Furthermore, they doped DTPPAO into HEA-AA photosensitive resin and prepared a series of complex 3D structures by 3D printing technology. Both UV irradiation curing time and humidity can quantitatively control the mechanical properties. The authors claimed that UV irradiation can manipulate the cross-linking degree of HEA-AA photosensitive resin and humidity can destroy intermolecular hydrogen bonding interactions, resulting in the tunable phosphorescence lifetimes. The research is meaningful. However, I don't think the results are exciting enough and the data are solid enough to support their conclusions. Therefore, I don't recommend its publication in Nature Communications. My major concerns are listed below.

- (1) The authors gave three reasons to support that their design is rational. However, the experimental data cannot support high PL efficiency and high afterglow PLQY. The PLQY values in table 1 are quite low.
- (2) Compared with reported phosphorescence molecules, the reviewer cannot agree that a twisted "U-shape" configuration matters much in the guest-host phosphorescence systems.
- (3) About molecular characterization, the authors merely offered ¹H NMR spectra and high-resolution mass spectrometry and claimed that possible interference of impurities was avoided in this work. It is not acceptable for a high-level journal. In addition, ¹H NMR spectra should extend to 0 ppm.
- (4) In figure S5, DSC curves show that the two target molecules may be not pure.
- (5) In DTPPAO/host systems, the phosphorescence band is centered at 503 nm. In DTPPAO/PMMA, the phosphorescence band is centered at 466 nm. A red shift of 37 nm occurs. A rational explanation should be given.
- (6) Line 160, the authors doped DTPPAO/ tBuDTPPAO into TPO, TPP and SF with a weight ratio of 1%. What's the influence of concentration? Is the phosphorescence property concentration dependent?
- (7) Figure S9, the authors offered the temperature-dependent steady-state PL spectra to evidence TADF and phosphorescence. However, the 442 nm band is greatly affected by fluorescence.
- (8) The authors declared that the triplet states of the host matrices show a synergistic effect to the excited states of the guest molecules. There are no experimental evidences to support it. Transient absorption spectrum may show the charge transfer from guest to host.
- (9) In the section of Theoretical calculations, the calculation results of TPO, TPP and SF should be provided.
- (10) "the moisture greatly breaks the hydrogen bonding interactions between DTPPAO and HEA-AA matrix." Direct evidence should be offered.

Reviewer #2 (Remarks to the Author):

This manuscript successfully demonstrated the first example of tunable afterglow materials for mechanical self-monitoring in complex 3D printing structures based on two new dual-emissive room temperature afterglow molecules. This is an interesting and attractive result. Besides, the photophysical properties and mechanism of tunable dual afterglow for compounds DTPPAO and tBuDTPPAO were well characterized and demonstrated. The mechanical properties of all the 3D structures can be well monitored by their afterglow properties, which can be quantitatively manipulated by either UV curing time or humidity. Overall, this work is a breakthrough for developing applications in real-time mechanical detection for 3D printed structures, and shows great promising potential in sensing, 3D and mechanical analysis. I strongly recommend it for publication in Nature Communications after a minor revision as listed below:

1. The sample preparation method of host-guest doping systems described in photophysical characterization section is melt-casting, the thermal analyses of all the host materials are also necessary.
2. As shown in Figure 1, the two new materials DTPPAO and tBuDTPPAO in doping systems show dual-emissive room temperature afterglow, referring to TADF and RTP mechanism. However, only the lifetimes of RTP are given, the lifetime decays of TADF band should be added.
3. The author mentioned that Förster and triplet-triplet Dexter energy transfer are involved for the tunable dual-emissive afterglow in the doping systems. But the photoluminescence and delayed spectra are not sufficient to prove the energy transfer process. It should be evaluated by the degree of the spectral overlap of the emission and absorption of the host and guest molecules, respectively. Further experiments are necessary to further prove the energy transfer process in the mechanism.
4. For the mechanical properties, tension experiments may help to understand the strength and ductility of the doping systems. I suggest to add this set of experiments.
5. As shown in Figure 5b, the author's schematic indicated that the table structure could be returned to its initial state by drying after water-soaking treatment, is it reversible? If it is, how is the fatigue resistance?

Reviewer #3 (Remarks to the Author):

In the text, the authors present two new fluorescent sensors for real-time monitoring of material changes. The authors demonstrate for the first time that the mechanical behaviours can be well monitored by controlling the UV irradiation curing time and humidity level and quantitatively manifested by afterglow lifetimes. The proposed structures are unknown, but their functionality is known. Moreover, even photoinitiators have been developed, allowing resins to cure, and their decomposition products act as probes (One-Component Cationic Photoinitiators from Tunable Benzylidene Scaffolds for 3D Printing Applications, *Macromolecules* 2021, 54, 15, 7070–7087). Therefore, in this respect, the article is not groundbreaking because there are works that describe the possibility of on line and in situ monitoring of the degree of polymer cross-linking during 3D printing.

Nevertheless, significant improvements are needed for the article to be accepted:

1) The compounds shown are built from a strong donor (DPA) and two acceptors, dibenzothiophene and phosphine oxide. Missing from the text is a reference, which would be a compound built with only DPA and dibenzothiophene, because on the posted results, the contribution of the phosphine group is ambiguous, e.g. the LUMO orbital in both cases is completely localized on dibenzothiophene.

2) In the methods section, the authors mention that C NMR spectra were recorded for the compounds; unfortunately, the supplement lacks these spectra and their descriptions. They should be added. Moreover, H and C NMR and mass spectra for intermediates I and II should also be added.

3) The language needs improvement. The text has numerous phrases, such as "easy processible process" (Page 2, line 51), which should be improved.

4) Page 3, lines 79-80 listed twice carbon fibres with two separate references. This should be corrected

5) Add to the supplement graphs the dependence of molar absorption coefficients on wavelength for both compounds

6) Photolysis diagrams of the presented compounds in solution at the wavelengths of light used in the study should be added to the supplement

After reading the manuscript I find it suitable for the journal but the authors are asked to make some clarifications and corrections in the major revision mode.

The responses to the reviewers' comments and corresponding corrections have been made as detailed below. Additional texts have been added and revised in the manuscript and Supporting Information, highlighted in yellow.

(Referees' comments are in black text, author responses are in blue.)

RESPONSE to REVIEWER COMMENTS:

To Reviewer #1:

In this manuscript, the authors report two novel D-A type emitters DTPPAO and *tBu*DTPPAO. When DTPPAO and *tBu*DTPPAO were doped into small molecules or polymer, tunable TADF-RTP can be achieved. The authors declared that the triplet states of the host matrices show a synergistic effect to the excited states of the guest molecules. Furthermore, they doped DTPPAO into HEA-AA photosensitive resin and prepared a series of complex 3D structures by 3D printing technology. Both UV irradiation curing time and humidity can quantitatively control the mechanical properties. The authors claimed that UV irradiation can manipulate the cross-linking degree of HEA-AA photosensitive resin and humidity can destroy intermolecular hydrogen bonding interactions, resulting in the tunable phosphorescence lifetimes. The research is meaningful. However, I don't think the results are exciting enough and the data are solid enough to support their conclusions. Therefore, I don't recommend its publication in Nature Communications. My major concerns are listed below.

We really appreciate the reviewer's invaluable comments and suggestions. In this manuscript, we reported the tunable afterglow properties (emission and lifetime) of two novel molecules DTPPAO and *tBu*DTPPAO doped in different hosts, and successfully achieved mechanical self-monitoring complex 3D printing structures. As far as we know, this is the first case demonstrating that the mechanical properties of 3D printing structures can be self-monitored by the afterglow properties of organic molecules, which have promising applications in real-time mechanical self-detection. To improve our manuscript according to the reviewer's comment, supplementary experiments were performed to improve our manuscript as follows.

(1) The authors gave three reasons to support that their design is rational. However, the experimental data cannot support high PL efficiency and high afterglow PLQY. The PLQY values in table 1 are quite low.

Response: We thank the reviewer for the comment. To ensure the accuracy, we have repeated the PLQY measurement of all the doping samples in ambient condition as summarized in Table R1 below. The data have also been updated in the manuscript (Table 1). *tBu*DTPPAO doped systems exhibit relatively higher PLQYs than DTPPAO systems. Despite the low PLQYs, the afterglow emission of the doped systems is quite strong compared with the delayed fluorescence band at room temperature, which can also be clearly observed with naked eyes, especially in SF host matrix (Figs. 1e-f). For instance, DTPPAO/SF and *tBu*DTPPAO/SF exhibit long afterglow lifetimes of 287.7 ms and 298.4 ms, respectively. The errors between the two measurements are within permission. In this work, we mainly focused on the tunable RTP properties and the application in mechanical self-monitoring, optimizing the quantum efficiency of the two compounds would be our next research task.

Table R1. PLQYs of DTPPAO and *tBu*DTPPAO in different hosts in air at room temperature.

PLQY (%)	TPO	TPP	SF
DTPPAO	4.01	5.60	5.07
tBu DTPPAO	9.15	12.62	8.38

(2) Compared with reported phosphorescence molecules, the reviewer cannot agree that a twisted “U-shape” configuration matters much in the guest-host phosphorescence systems.

Response: We thank the reviewer’s comment. We did not describe this adequately. The molecular design strategy in this work is to achieve intramolecular through bond charge transfer between the donor DPA and weak acceptor DBT, and intramolecular through space charge transfer between DPA and the stronger acceptor DPO units. This can be confirmed by frontier molecular orbitals of DTPPAO and *tBu*DTPPAO from density function theory (DFT) calculations as shown in Figs. S54-55. In both compounds, HOMO is localized on the donor DPA unit with a slight extension to the adjacent phenyl ring of DBT unit, LUMO is mainly localized on the DBT unit, while LUMO+1 is localized on the DPO unit. This distribution is mainly due to the degeneracy of energy level [1,2] that LUMO+1 shows a similar energy level as LUMO, which provides an electron transfer channel from HOMO to LUMO+1. Therefore, both acceptor units (DBT and DPO) contribute to the charge transfer process with this molecular configuration. In addition, strong intermolecular π - π stacking in the doping systems can also enhance the afterglow property.

In addition, we have also investigated the afterglow properties of their analogue D-A molecule (DTPA), which can further confirm the role of the “U-shape” configuration of D-A-A’ molecules. Fig. R1a (Fig. S36a) shows the chemical structure of DTPA, its HOMO/LUMO distributions are similar to those of DTPPAO and *tBu*DTPPAO. The redshifted PL spectra in solution with different polarities indicate its singlet state of charge transfer character (Fig. R1b (Fig. S36b)). Here triphenylphosphine (TPP) and diphenyl-sulfone (SF) were chosen as the host matrices. As shown in Fig. R2 (Fig. S37), DTPA/SF doping system shows an obvious RTP emission shoulder in the steady-state emission while RTP emission in DTPA/TPP is relatively weak which is covered by its broad PL emission tail, peaking at 504 nm. The phosphorescence lifetimes of DTPA/TPP and DTPA/SF systems are 4.0 and 149.8 ms, respectively (Fig. R3 (Fig. S38)). In contrast, obvious RTP emission and longer lifetimes of hundreds of milliseconds in all doping systems were achieved in D-A-A’ compounds with the “U-shape” configuration, which further demonstrate the significance of the additional DPO acceptor unit in DTPPAO and *tBu*DTPPAO.

Following the reviewer’s suggestion, the following texts have been added in the manuscript on Page 4: “LUMOs+1 mainly occupy the relatively strong donor DPO. This is due to the degeneracy of energy level, which provides an additional electron transfer channel from DPA to DPO unit. Therefore, both acceptor units (DBT and DPO) can contribute to the charge transfer process with this molecular configuration.” And “The long-lived afterglow in the guest/host system gives further evidence showing the rationality of our molecular design strategy, as its analogue D-A (DTPA) without an additional DPO acceptor unit shows much shorter RTP lifetimes in host matrices, which are 4.0 ms in DTPA/TPP and 149.8 ms in DTPA/SF (Figs. S36-38).” was added on Page 7.

Fig. R1 (Fig. S36) a) Molecular structure of DTPA and its HOMO/LUMO distributions calculated at the B3LYP/6-311G(d) level. b) Normalized UV-vis absorption and emission spectra in n-Hexane and dichloromethane (DCM) solutions (1×10^{-5} M).

Fig. R2 (Fig. S37) a) Steady-state emission and b) delayed spectra (delay time of 1 ms) of DTPA/TPP and DTPA/SF systems (1 wt%) at ambient condition.

Fig. R3 (Fig. S38) Time-resolved decay curves of DTPA/TPP and DTPA/SF doping systems (1 wt%) at the phosphorescence band of 500 nm at ambient condition.

References

1. Herrick D R. Degeneracies in energy levels of quantum systems of variable dimensionality. *J. Math. Phys.* **16**(2), 281-283 (1975).
 2. Kubo H, Hirose T and Matsuda K. Control over the emission properties of [5]helicenes based on the symmetry and energy levels of their molecular orbitals, *Org. Lett.* **19**, 1776–1779 (2017).
- (3) About molecular characterization, the authors merely offered ¹H NMR spectra and high-resolution mass spectrometry and claimed that possible interference of impurities was avoided in this work. It is not acceptable for a high-level journal. In addition, ¹H NMR spectra should extend to 0 ppm.

Response: We sincerely appreciate the reviewer's comment to improve our manuscript. According to the reviewer's suggestion, we have added and updated the ¹H NMR, ¹³C NMR, high resolution mass spectra, high performance liquid chromatography (HPLC) spectra and element analyses (EA) of DTPPAO and *t*BuDTPPAO as shown in Figs. R4-8 and Table R2 (Figs. S1-19 and Table S1). All the characterizations confirm the purity of our compounds DTPPAO and *t*BuDTPPAO.

Fig. R4 (Fig. S3) ¹H NMR spectrum of DTPPAO.

Fig. R5 (Fig. S6) ¹H NMR spectrum of *t*BuDTPPAO.

Fig. R6 (Fig. S9) ¹³C NMR spectrum of DTPPAO.

Fig. R7 (Fig. S12) ^{13}C NMR spectrum of *tBuDTPPAO*.

Fig. R8 (Fig. S19) High performance liquid chromatography (HPLC) spectra of DTPPAO and *tBuDTPPAO* in methanol solution.

Table R2 (Table S1). Elemental analyses (EA) of DTPPAO and *tBuDTPPAO*.

Element content (%)		C	H	N
DTPPAO	Theoretical value	78.38	4.75	2.54
	Experimental value	78.553	4.695	2.543
t BuDTPPAO	Theoretical value	79.61	6.38	2.11
	Experimental value	79.875	6.409	2.043

(4) In figure S5, DSC curves show that the two target molecules may be not pure.

Response: We thank the reviewer for the comment. We have checked the original data, the measurement results were not clear enough and misleading. To confirm the purity of our target molecules, we have repeated the DSC measurements of DTPPAO and *t*BuDTPPAO as shown in Fig. R9 (Fig. S20). The melting point (T_m) of DTPPAO and *t*BuDTPPAO were obtained from the maximum heat flow. Both of them exhibit relatively weak heat flow, which is mainly due to their poor crystallinity that can be confirmed by the X-ray powder diffraction [3] (Fig. R10 (Fig. S21)). The ^1H NMR spectra, ^{13}C NMR spectra, high performance liquid chromatography (HPLC) and elemental analysis (EA) also confirmed the purity of DTPPAO and *t*BuDTPPAO as shown in the response to the third comment (Figs. R4-8).

Fig. R9 (Fig. S20) DSC curves of DTPPAO and *t*BuDTPPAO under N₂ atmosphere at 10°C/min heating rate.

Fig. R10 (Fig. S21) X-ray powder diffraction of DTPPAO and *t*BuDTPPAO.

References

- Li, Y., Zheng, N., Yu, L., Wen, S., Gao, C., Sun, M. and Yang, R. A simple phenyl group introduced at the tail of alkyl side chains of small molecular acceptors: new strategy to balance the crystallinity of acceptors and miscibility of bulk heterojunction enabling highly efficient organic solar cells. *Adv. Mater.* **31**, 1807832 (2019).

(5) In DTPPAO/host systems, the phosphorescence band is centered at 503 nm. In DTPPAO/PMMA, the phosphorescence band is centered at 466 nm. A red shift of 37 nm occurs. A rational explanation should be given.

Response: We thank the reviewer for the comment. A rational explanation has been added by the comparison of all phosphorescence spectra. As DTPPAO shows a similar phosphorescence spectrum in different doping systems, here we take DTPPAO/TPO as an example to compare the phosphorescence spectrum at 77 K and room temperature with that in PMMA matrix at 77 K as shown in Fig. R11. DTPPAO/TPO exhibits a slightly red-shifted phosphorescence spectrum peaking around 473 nm at 77 K, compared with that of DTPPAO/PMMA film (466 nm). One reason is the influence of the host polarity [4]. PMMA is a nonpolar polymer matrix, which mainly provides a rigid environment for the suppression of non-radiative decays. While the

small molecule matrix can increase the polarity of the doping systems, resulting in a spectral redshift. In addition, the sufficient intermolecular interaction between DTPPAO and TPO is another significant reason for the slight redshift [5]. Moreover, different from the phosphorescence at 77 K, DTPPAO/TPO exhibits a more structural spectrum at room temperature (478/503 nm). This may be because the low temperature suppresses the nonradiative pathways to a great extent, resulting in a contribution from higher triplet states. Whereas at room temperature, the phosphorescence derives from the T_1 state that has a localized excited character. All these reasons account for the spectral shift of phosphorescence synergistically.

Following the reviewer's suggestion, the following sentences were added on page 7: "The phosphorescence spectra of guest molecules in small organic matrices exhibit a slight redshift compared with that in PMMA film. One reason could be the influence of the host polarity. In addition to the increased rigidity that suppresses the non-radiative pathways, the small molecule matrices (TPO, TPP and SF) can also increase the polarity of the doping systems, resulting in a spectral redshift. Moreover, the sufficient intermolecular interactions between guest and small molecule hosts can also induce the spectral redshift. All the possible reasons account for the spectral shift of phosphorescence synergistically."

Fig. R11 Phosphorescence spectra of DTPPAO in PMMA and TPO matrices recorded at delay time of 8 ms at room temperature (RT) and 77 K.

References

4. Liang, X., Zheng, Y.-X., and Zuo, J.-L. Two-photon ionization induced stable white organic long persistent luminescence. *Angew. Chem. Int. Ed.* **60**, 16984-16988 (2021).
5. Cai, S., Shi, H., Li, J., Gu, L., Ni, Y., Cheng, Z., Wang, S., Xiong, W.-w., Li, L., An, Z. and Huang, W., Visible-light-excited ultralong organic phosphorescence by manipulating intermolecular interactions. *Adv. Mater.* **29**, 1701244 (2017).

(6) Line 160, the authors doped DTPPAO/tBuDTPPAO into TPO, TPP and SF with a weight ratio of 1%. What's the influence of concentration? Is the phosphorescence property concentration dependent?

Response: We thank the reviewer for this invaluable comment. According to the suggestion, we have performed the concentration-dependent measurements of DTPPAO and *tBu*DTPPAO doping systems, including steady-state photoluminescence (PL) spectra and time-resolved decay curves as shown in Figs. R12-17 (Figs. S47-52). With a relatively large triplet-triplet energy gap, DTPPAO/TPO system exhibits a weak phosphorescence emission covered by the PL spectra and a short phosphorescence lifetime. While in SF matrix, obvious phosphorescence spectra and longer lifetimes are observed even at doping concentration of 0.1%. The relative intensity of PL emission (TADF and phosphorescence) of DTPPAO and *tBu*DTPPAO doping systems exhibits an enhancement with increasing doping concentration from 0.1% to 1%, the time-resolved decays (recorded at 500 nm in DTPPAO/host and 490 nm in *tBu*DTPPAO/host systems) decrease correspondingly. It is proposed that triplet-triplet energy transfer becomes more efficient at higher doping concentrations. Therefore, we demonstrate that the phosphorescence properties of these doping systems are dependent on the concentration that stronger RTP emission and shorter RTP lifetime can be obtained at higher doping concentrations.

Following the reviewer's suggestion, the following sentences were added on page 8: "As shown in Figs. S47-52, all the doping systems of DTPPAO and *tBu*DTPPAO exhibit a trend that phosphorescence emission enhances (500 nm in DTPPAO/SF and 490 nm in *tBu*DTPPAO/SF) with increasing doping concentration, while the time-resolved decay decreases correspondingly. It indicates that energy transfer becomes more efficient at a higher doping concentration."

Fig. R12 (Fig. S47) a) Steady-state photoluminescence spectra and b) time-resolved decay curves recorded at 500 nm of DTPPAO/TPO doping system at different concentrations.

Fig. R13 (Fig. S48) a) Steady-state photoluminescence spectra and b) time-resolved decay curves recorded at 500 nm of DTPPAO/TPP doping system at different concentrations.

Fig. R14 (Fig. S49) a) Steady-state photoluminescence spectra and b) time-resolved decay curves recorded at 500 nm of DTPPAO/SF doping system at different concentrations.

Fig. R15 (Fig. S50) a) Steady-state photoluminescence spectra and b) time-resolved decay curves recorded at 490 nm of *t*BuDTPPAO/TPO doping system at different concentrations.

Fig. R16 (Fig. S51) a) Steady-state photoluminescence spectra and b) time-resolved decay curves recorded at 490 nm of *tBuDTPPAO/TPP* doping system at different concentrations.

Fig. R17 (Fig. S52) a) Steady-state photoluminescence spectra and b) time-resolved decay curves recorded at 490 nm of *tBuDTPPAO/SF* doping system at different concentrations.

(7) Figure S9, the authors offered the temperature-dependent steady-state PL spectra to evidence TADF and phosphorescence. However, the 442 nm band is greatly affected by fluorescence.

Response: We thank the reviewer for the comment. This explanation was not clear in the original manuscript. We claimed it as the TADF mechanism as the intensity of fluorescence band (~442 nm) shows a positive temperature dependence, which increases with the increasing temperature, indicating a thermally activated process (cf. Fig. S29). In addition, we

have added the time-resolved decay curves of DTPPAO/host recorded at the fluorescence band (425 nm) as shown in Fig. R18 (Fig. S30). The lifetimes of all three doping systems are in millisecond range (6.05 ms in TPO, 6.27 ms in TPP and 21.99 ms in SF), further confirming its persistent TADF mechanism. Likewise, *tBu*DTPPAO/host systems also exhibit long lifetimes in milliseconds (Fig. R19 (Fig. S35)).

Following the reviewer's suggestion, we have added a sentence in Page 6: "The millisecond range of the fluorescence band further confirms its TADF process (Fig. S30)."

Fig. R18 (Fig. S30) Time-resolved decay curves of DTPPAO/host doping systems recorded at the fluorescence band of 425 nm at room temperature.

Fig. R19 (Fig. S35) Time-resolved decay curves of *tBuDTPPAO*/host doping systems recorded at 430 nm at room temperature.

(8) The authors declared that the triplet states of the host matrices show a synergistic effect to the excited states of the guest molecules. There are no experimental evidences to support it. Transient absorption spectrum may show the charge transfer from guest to host.

Response: We sincerely thank the reviewer's invaluable suggestion to improve our manuscript. We agree that it is important to explain the synergistic effect of the triplet excited states between host and guest molecules in details. According to the reviewer's suggestion, we have investigated the excited-state dynamics of DTPPA/SF doping system using femtosecond transient absorption (fs-TA) spectroscopy to further verify the triplet-triplet energy transfer [6-8]. All the samples were measured using solid-state films upon excitation at 350 nm in ambient condition, which may induce complex and broad signal in their respective TA spectra. As shown in Fig. R20a (Fig. S45a), SF shows two broad positive absorption bands. Over time, a band at around 637 nm exhibits a clear intensity increase from 139.77 ps, which can be assigned to the excited-state absorption (ESA) of $T_1 \rightarrow T_n$. Compound DTPPAO also shows a broad TA band from 540 to 700 nm after excitation (Fig. R20b (Fig. S45b)). Upon longer delay time around 756.43 ps, two distinctive ESA bands peaking at 575 and 620 nm dominates. Both bands can be assigned to the triplet ESA of DTPPAO as they grow continuously in intensity and can live up to 2464.30 ps. As shown in Fig. R21 (Fig. S46), DTPPAO/SF doping system exhibits a broad positive TA band, including contributions from both DTPPAO and SF molecules, which may make the TA spectra more complex. A clear spectral redshift from 642 nm to around 600 nm

is observed, corresponding to the triplet ESA bands of SF and DTPPAO, respectively. With increasing delay time, the intensity of 642 nm band from SF decreases gradually, while the broad band (567 to 616 nm) from DTPPAO increases in intensity which appears earlier (~509.93 ps) than the pure DTPPAO film and lives up to a few nanoseconds. The intensity changes and gradual structured TA spectra of DTPPAO/SF provides qualitative evidence for the occurring of distinct triplet-triplet energy transfer between host and guest molecules.

Following the reviewer's suggestion, we added the following sentences in the revised manuscript on Page 8: "To further verify the triplet-triplet energy transfer in the guest/host doping systems, the excited state dynamics of DTPPAO/SF system were investigated using femtosecond transient absorption (fs-TA) spectroscopy. All the samples were measured using solid-state films upon excitation at 350 nm in ambient conditions, which may induce complex and broad signal in their respective TA spectra. Both SF and DTPPAO exhibit a broad positive absorption band in a large wavelength range. Upon longer delay time, the band at around 637 nm in SF shows a clear intensity increase from 139.77 ps, which can be assigned to the excited-state absorption (ESA) of $T_1 \rightarrow T_n$ (Fig. S45a). Two distinctive ESA bands peaking at 575 and 620 nm appeared at 756.43 ps are identified as the triplet ESA bands of DTPPAO as they grow continuously in intensity and can live up to 2464.30 ps (Fig. S45b). As shown in Fig. S46, DTPPAO/SF doping system exhibits a broad positive TA band, including contributions from both DTPPAO and SF molecules. A clear spectral redshift from 642 nm to around 600 nm is observed, corresponding to the triplet ESA bands of SF and DTPPAO, respectively. With increasing delay time, the intensity of 642 nm band from SF decreases gradually, while the broad band (567-616 nm) from DTPPAO increases in intensity which appears earlier (~509.93 ps) than the pure DTPPAO film and lives up to a few nanoseconds. The intensity changes and gradual structured TA spectra of DTPPAO/SF provides qualitative evidence for the occurring of distinct triplet-triplet energy transfer between host and guest molecules."

Fig. R20 (Fig. S45) Femtosecond transient absorption (fs-TA) spectra of a) SF and b) DTPPAO at different delay times upon excitation at 350 nm (The peak around 700 nm marked as * is caused by the instrument).

Fig. R21 (Fig. S46) a) Femtosecond transient absorption (fs-TA) spectra of 5 wt%DTPPAO/SF doping system at different delay times upon excitation at 350 nm.

References

- W. Hu, T. He, H. Zhao, H. Tao, R. Chen, L. Jin, J. Li, Q. Fan, W. Huang, A. Baev, P. N. Prasad, Stimuli-responsive reversible switching of intersystem crossing in pure organic material for smart photodynamic therapy. *Angew. Chem. Int. Ed.* **58**, 11105-11111 (2019).

7. Lai, R., Liu, Y., Luo, X. et al. Shallow distance-dependent triplet energy migration mediated by endothermic charge-transfer. *Nat. Commun.* **12**, 1532 (2021).
8. Gehrig, D., Howard, I. A., Kamm, V., Dyer-Smith, C., Etzold, F. and Laquai, F. Charge generation in polymer: perylene diimide blends probed by Vis-NIR broadband transient absorption pump-probe spectroscopy. *Proc. SPIE*, **8811**, 110-119 (2013).

(9) In the section of Theoretical calculations, the calculation results of TPO, TPP and SF should be provided.

Response: We sincerely thank the reviewer's comment to improve our manuscript. According to the reviewer's suggestion, we have calculated the frontier molecular orbitals of TPO, TPP and SF by density functional theoretical (DFT) calculations based on the optimized molecular geometries in gas phase at the B3LYP/6-311G(d) level [9]. All computations were performed using the Gaussian 09 package.[10,11] As shown in Fig. R22 (Fig. S53), the highest occupied molecular orbitals (HOMOs) and the lowest unoccupied molecular orbitals (LUMOs) of TPO, TPP and SF matrices are all localized on their whole molecular skeleton, revealing their localized excited character. And all the three host matrices exhibit large energy gaps (~5.9 eV in TPO and TPP, ~5.3 eV in SF), indicating their potential as host matrices.

Following the reviewer's suggestion, we added the following sentences in the revised manuscript on Page 9: "The HOMOs and LUMOs of TPO, TPP and SF matrices are all localized on their whole molecular skeleton (Fig. S53)."

Fig. R22 (Fig. S53) Frontier molecular orbitals based on optimized geometries of TPO, TPP and SF with B3LYP/6-311G(d) functionals.

References

- Lu, T. & Chen, F. Multiwfn: A multifunctional wavefunction analyzer. *J Comput. Chem.* **33**, 580–592 (2012).
- Gao, F. et al. High-efficiency blue thermally activated delayed fluorescence from donor-acceptor-donor systems: Via the through-space conjugation effect. *Chem. Sci.* **10**, 5556–5567 (2019).
- Gaussian 09, Revision A.02, M. J. Frisch, G. W. Trucks, H. B. Schlegel, G. E. Scuseria, M. A. Robb, J. R. Cheeseman, G. Scalmani, V. Barone, G. A. Petersson, H. Nakatsuji, X. Li, M. Caricato, A. Marenich, J. Bloino, B. G. Janesko, R. Gomperts, B. Mennucci, H. P. Hratchian, J. V. Ortiz, A. F. Izmaylov, J. L. Sonnenberg, D. Williams-Young, F. Ding, F. Lipparini, F. Egidi, J. Goings, B. Peng, A. Petrone, T. Henderson, D. Ranasinghe, V. G. Zakrzewski, J. Gao, N. Rega, G. Zheng, W. Liang, M. Hada, M. Ehara, K. Toyota, R. Fukuda, J. Hasegawa, M. Ishida, T. Nakajima, Y. Honda, O. Kitao, H. Nakai, T. Vreven, K. Throssell, J. A. Montgomery, Jr., J. E. Peralta, F. Ogliaro, M. Bearpark, J. J. Heyd, E. Brothers, K. N. Kudin, V. N. Staroverov, T. Keith, R. Kobayashi, J. Normand, K. Raghavachari, A. Rendell, J. C. Burant, S. S. Iyengar, J. Tomasi,

M. Cossi, J. M. Millam, M. Klene, C. Adamo, R. Cammi, J. W. Ochterski, R. L. Martin, K. Morokuma, O. Farkas, J. B. Foresman, and D. J. Fox, Gaussian, Inc., Wallingford CT, 2016.

(10) "the moisture greatly breaks the hydrogen bonding interactions between DTPPAO and HEA-AA matrix." Direct evidence should be offered.

Response: Thanks for the reviewer's invaluable suggestion. We are sorry for inaccurate description and inadequate results for the hydrogen bonding interactions in DTPPAO/HEA-AA doping film. We know that the afterglow emission is generally not from an isolated species, but molecular assemblies, such as intermolecular hydrogen bonding.[12,13] Two types of hydrogen-bonding interactions are concurrent in the DTPPAO/HEA-AA system that can jointly restrict the molecular vibrations, i.e., hydrogen bonds between matrix polymers, and between DTPPAO and HEA-AA. HEA-AA chain-chain hydrogen bonds can suppress the diffusional motion of the matrix and DTPPAO/HEA-AA hydrogen bonds can reinforce the restriction of vibration of DTPPAO.

We have recorded the Fourier-transform infrared (FTIR) spectra of DTPPAO/HEA-AA film to give more direct evidence understanding the hydrogen bonding in this system. As shown in Fig. R23 (Fig. S70), the dry film exhibits noticeable C=O and O-H bonds peaking at 1713 cm^{-1} and 3456 cm^{-1} , respectively, which can be attributed to the associated hydroxyl group between adjacent DTPPAO and HEA-AA. When the film was fumed with water vapor, the O-H peak shape becomes wider and the peak shifts to short wavenumber of 3427 cm^{-1} in ambient condition and further 3396 cm^{-1} in humidity condition. C=O peak also shows a gradual shift to 1710 cm^{-1} and 1706 cm^{-1} in ambient and humidity condition, respectively. This reveals that the presence of water in DTPPAO/HEA-AA could increase the association degree of hydroxyl groups. Combined with the changes in the afterglow properties, we demonstrate that the presence of water weakens and breaks the hydrogen bonding interactions between adjacent DTPPAO and HEA-AA, resulting in a significant activation of vibrational dissipation and consequent phosphorescence quenching.[14] Furthermore, hydrogen bonding interactions can also be confirmed by the comparison of the afterglow properties between DTPPAO/PMMA and DTPPAO/HEA-AA films. DTPPAO/HEA-AA exhibits an enhanced RTP intensity and much longer afterglow lifetime ($\sim 500\text{ ms}$) as the OH-groups in HEA-AA are from the nature of its own structure while they are mainly from the tautomerization of the methyl ester groups in PMMA.[15] Thus, strong hydrogen bonding interactions can be formed in DTPPAO/HEA-AA system instead of DTPPAO/PMMA, indicating the existence of hydrogen bonding. In addition, we have observed that the RTP intensity of the 500 nm emission band gradually decreases with

the increasing water-soaking time, whereas the fluorescence emission peak at 435 nm remains unchanged as shown in Fig. 3c. Meanwhile, the lifetime of the RTP band decreases gradually from 489.1 ms in dry condition to 2.9 ms after water vapor for 24 hours in a 58% moisture supersaturated NaBr solution (Fig. 3d and Figs. S64-69). The photographs in Fig. S65 also depict that the afterglow of DTPPAO/HEA-AA film almost vanishes in 58%RH environment for 24 hours.

Following the reviewer's suggestion, we added the following sentences in the revised manuscript on Page 11: "This can be confirmed by the Fourier-transform infrared (FTIR) spectra as shown in Fig. S70. The dry DTPPAO/HEA-AA film exhibits noticeable C=O and O-H bonds peaking at 1713 and 3456 cm^{-1} , respectively, which can be attributed to the associated hydroxyl group between adjacent DTPPAO and HEA-AA. After fuming with water vapor, the C=O and O-H peaks shift to 1706 and 3396 cm^{-1} , respectively, indicating an increase in the association degree of hydroxyl groups. Combined with the changes in afterglow properties, we demonstrate that the presence of water weakens and breaks the hydrogen bonding interactions between adjacent DTPPAO and HEA-AA, resulting in a significant activation of vibrational dissipation and consequent phosphorescence quenching."

Fig. R23 (Fig. S70) Fourier transform infrared (FTIR) spectra in DTPPAO/HEA-AA film under the heating and water fuming stimuli (dry: drying treatment at oven; room: at ambient condition for 24 h; humidity: at 58%RH moisture supersaturated NaBr solution for 24 h).

References

12. Kwon, M.S., Lee, D., Seo, S., Jung, J. and Kim, J. Tailoring Intermolecular Interactions for Efficient Room-Temperature Phosphorescence from Purely Organic Materials in Amorphous Polymer Matrices. *Angew. Chem. Int. Ed.*, **53**, 11177-11181 (2014).
13. Li, D., Yang, Y., Yang, J. et al. Completely aqueous processable stimulus responsive organic room temperature phosphorescence materials with tunable afterglow color. *Nat Commun* **13**, 347 (2022).
14. Wu, H., Gu, L., Baryshnikov, G. V., Wang, H., Minaev, B. F., Ågren, H. and Zhao, Y. Molecular phosphorescence in polymer matrix with reversible sensitivity. *ACS Applied Materials & Interfaces*, **12**, 20765-20774 (2020).
15. Thomas, H., Pastoetter, D. L., Gmelch, M., Achenbach, T., Schlögl, A., Louis, M., Feng, X. and Reineke, S. Aromatic phosphonates: a novel group of emitters showing blue ultralong room temperature phosphorescence. *Adv. Mater.* **32**, 2000880 (2020).

Reviewer #2:

This manuscript successfully demonstrated the first example of tunable afterglow materials for mechanical self-monitoring in complex 3D printing structures based on two new dual-emissive room temperature afterglow molecules. This is an interesting and attractive result. Besides, the photophysical properties and mechanism of tunable dual afterglow for compounds DTPPAO and *tBu*DTPPAO were well characterized and demonstrated. The mechanical properties of all the 3D structures can be well monitored by their afterglow properties, which can be quantitatively manipulated by either UV curing time or humidity. Overall, this work is a breakthrough for developing applications in real-time mechanical detection for 3D printed structures, and shows great promising potential in sensing, 3D and mechanical analysis. I strongly recommend it for publication in Nature Communications after a minor revision as listed below:

We thank the reviewer for the supportive comments and for highlighting the breakthrough of our work in developing mechanical self-monitoring 3D printed structures.

1. The sample preparation method of host-guest doping systems described in photophysical characterization section is melt-casting, the thermal analyses of all the host materials are also necessary.

Response: We appreciate the reviewer's helpful suggestion. Thermal analyses are indeed important data to evaluate molecular stability and purity, the melting points are also key

parameters for sample preparation by the melt-casting method. We have performed the DSC measurements of two guest compounds (DTPPAO and *tBu*DTPPAO), three host molecules (TPO, TPP and SF) and host/guest doping systems for a better demonstration as shown in Figs. R24-26 (Fig. S22-24). The DSC curves of DTPPAO and *tBu*DTPPAO are presented in Fig. R9 (Fig. S20) as shown in the response to the fourth comment of the first reviewer. The corresponding data are summarized in Table R3 (Table S2). All materials exhibit high purities and thermal analysis for the study, and the feasibility of sample preparation via melt-casting method. In addition, the guest/host doping systems show different melting points varied with the host materials.

To improve the paper, we have added sentences on page 6: "The DSC (differential scanning calorimeter) and TGA (thermogravimetry analyses) studies indicate that the melting points of the guest/host systems varied with the host materials (Figs. S23-24)."

Fig. R24 (Fig. S22) Thermogravimetry analyses (TGA) curves of guest (DTPPAO and *tBu*DTPPAO) and host matrices (TPO, TPP and SF) under N₂ atmosphere at 20°C/min heating rate.

Fig. R25 (Fig. S23). Differential scanning calorimeter (DSC) curves of host molecules (TPO, TPP and SF) under N₂ atmosphere at 10°C/min heating rate.

Fig. R26 (Fig. S24). Differential scanning calorimeter (DSC) curves of a) DTTPPAO/host and b) tBuDTTPPAO/host doping systems under N₂ atmosphere at 10°C/min heating rate.

Table R3 (Table S2). Thermal properties of crystalline powders for guest (DTTPPAO and tBuDTTPPAO), host materials (TPO, TPP and SF) and host/guest doping systems.

	DTPPAO	t BuDTPPAO	TPO	TPP	SF
T_m (°C)	94.0	95.9	160.8	83.5	129.5
T_d (°C)	388.2	388.3	239.5	202.2	213.6

*melting temperature (T_m) and decomposition temperatures (T_d) of 5% weight loss under N_2 atmosphere at 10°C/min and 20°C/min heating rate.

2. As shown in Figure 1, the two new materials DTPPAO and *t*BuDTPPAO in doping systems show dual-emissive room temperature afterglow, referring to TADF and RTP mechanism. However, only the lifetimes of RTP are given, the lifetime decays of TADF band should be added.

Response: Thanks for the reviewer's invaluable comment to improve our manuscript. According to the reviewer's suggestion, we have added the time-resolved decay curves of DTPPAO and *t*BuDTPPAO in different host matrices recorded at the TADF bands of 425 and 430 nm, respectively, as shown in Figs. R18-19 (Figs. S30 and 35) in the response to the 7th comment of the first reviewer. All doping systems of DTPPAO and *t*BuDTPPAO show long TADF lifetimes in a millisecond range.

Fig. R18 (Fig. S30) Time-resolved decay curves of DTPPAO/host doping systems recorded at 425 nm at room temperature.

Fig. R19 (Fig. S35) Time-resolved decay curves of *tBuDTPPAO*/host doping systems recorded at 430 nm at room temperature.

3. The author mentioned that Förster and triplet-triplet Dexter energy transfer are involved for the tunable dual-emissive afterglow in the doping systems. But the photoluminescence and delayed spectra are not sufficient to prove the energy transfer process. It should be evaluated by the degree of the spectral overlap of the emission and absorption of the host and guest molecules, respectively. Further experiments are necessary to further prove the energy transfer process in the mechanism.

Response: Thanks for the reviewer's invaluable comment. the reviewer's suggestion, the absorption and emission spectra of the host and guest molecules were conducted. The large overlap between the fluorescence spectra of host matrices and the absorption spectra of DTPPAO (*tBuDTPPAO*) reveals the Förster energy transfer process from host to guest (Fig. R27). As Dexter energy transfer involves electron exchange and typically occurs within a short-range (<10 Å)[16], the concentration dependent measurements including PL spectra and time-resolved decay curves were performed. As shown in Figs. S47-52, all the doping systems of DTPPAO and *tBuDTPPAO* exhibit an obvious intensity enhancement of the RTP band (500 nm in DTPPAO/SF and 490 nm in *tBuDTPPAO*/SF) with increasing doping concentration, and the time-resolved decay decreases correspondingly. It reveals that a higher energy transfer efficiency can be obtained as doping concentration increases, confirming the Dexter-type triplet-triplet energy transfer.[17]

Furthermore, the excited-state dynamics of DTPPA/SF doping system using femtosecond transient absorption (fs-TA) spectroscopy[6-8] were investigated to further verify the triplet-triplet energy transfer. As shown in Fig. R21 (Fig. S46), DTPPAO/SF doping system exhibits a broad positive TA band, including contributions from both DTPPAO and SF molecules, which may make the TA spectra more complex. A clear spectral redshift from 642 nm to around 600 nm is observed, corresponding to the triplet ESA bands of SF and DTPPAO, respectively. With increasing delay time, the intensity of 642 nm band from SF decreases gradually, while the broad band (567 to 616 nm) from DTPPAO increases in intensity which appears earlier (~509.93 ps) than the pure DTPPAO film and lives up to a few nanoseconds. The intensity changes and gradual structured TA spectra of DTPPAO/SF provides qualitative evidence for the occurring of distinct triplet-triplet energy transfer between host and guest molecules. Detailed experimental results are shown in the response to the 6th comment of the first reviewer.

Fig. R27 Normalized absorption spectra of DTPPAO and *tBu*DTPPAO, and steady-state photoluminescence spectra of host matrices (TPO, TPP and SF).

References

16. Skourtis, S. S., Liu, C., Antoniou, P., Virshup, A. M. & Beratan, D. N. Dexter energy transfer pathways. *Proc. Natl. Acad. Sci. U. S. A.* **113**, 8115-8120 (2016).
17. Wang, J.-X. et al. Organic composite crystal with persistent room-temperature luminescence above 650 nm by combining triplet-triplet energy transfer with thermally activated delayed fluorescence. *CCS Chem.* **2**, 1391-1398 (2020).

4. For the mechanical properties, tension experiments may help to understand the strength and ductility of the doping systems. I suggest to add this set of experiments.

Response: Thanks for the reviewer's helpful comment. According to the reviewer's suggestion, we performed mechanical tensile measurements on DTPPAO/HEA-AA film at different photocuring and water-soaking times. As we can see from Figs. R28-29 (Figs. S73-74), the Young's modulus of DTPPAO/HEA-AA film increases gradually as photocuring time increases from 10 s to 18000 s. Moreover, with increasing water-soaking times, the Young's modulus of the DTPPAO/HEA-AA film decreases from initial 575.7 MPa to about 1.2 MPa (Figs. R30-31 (Figs. S77-78)). All the repeated experiments are consistent with the trend of Fig. 4a in the original manuscript and demonstrate the strength and ductility of DTPPAO/HEA-AA system.

Fig. R28 (Fig. S73) Mechanical tensile curves of DTPPAO/HEA-AA at different photocuring time (10, 60, 300, 1800, 18000 s).

Fig. R29 (Fig. S74) Young's modulus of DTPPAO/HEA-AA at different photocuring time (10, 60, 300, 1800, 18000 s).

Fig. R30 (Fig. S77) Mechanical tensile curves of DTPPAO/HEA-AA (photocuring time: 5 h) with treatment in humid environment (58%RH) at different times (0, 1, 2, 4, 8, 12, 24 h).

Fig. R31 (Fig. S78) Young's modulus of DTPPAO/HEA-AA (photocuring time: 5 h) with treatment in humid environment (58%RH) for different times (0, 1, 2, 4, 8, 12, 24 h).

5. As shown in Figure 5b, the author's schematic indicated that the table structure could be returned to its initial state by drying after water-soaking treatment, is it reversible? If it is, how is the fatigue resistance?

Response: Thanks for the reviewer's helpful suggestion. We have placed the "table" structure (DTPPAO/HEA-AA) in alternating supersaturated NaBr solution (58RH%) and oven to change the moisture content and investigate its fatigue resistance. As shown in Fig. R32 (Fig. S79), the phosphorescence lifetime of DTPPAO/HEA-AA decreases from about 450 ms in dry condition to almost 0 ms under moisture treatment for 24 h. The structure presents extremely stable stability after five repeated cycles of water-absorbing and drying treatments.

Following the reviewer's suggestion, we have added a sentence on Page 15: "Moreover, the structure exhibits excellent fatigue resistance, as its phosphorescence lifetime shows a reversible change after five repeated cycles of water-absorbing and drying treatments (Fig. S79)."

Fig. R32 (Fig. S79) The phosphorescence lifetimes of DTPPAO/HEA-AA after repeated cycles of water absorbing and drying processes.

Reviewer #3 (Remarks to the Author):

In the text, the authors present two new fluorescent sensors for real-time monitoring of material changes. The authors demonstrate for the first time that the mechanical behaviours can be well monitored by controlling the UV irradiation curing time and humidity level and quantitatively manifested by afterglow lifetimes. The proposed structures are unknown, but their functionality is known. Moreover, even photoinitiators have been developed, allowing resins to cure, and their decomposition products act as probes (One-Component Cationic Photoinitiators from Tunable Benzylidene Scaffolds for 3D Printing Applications, *Macromolecules* 2021, 54, 15, 7070–7087). Therefore, in this respect, the article is not groundbreaking because there are works that describe the possibility of on line and in situ monitoring of the degree of polymer cross-linking during 3D printing. Nevertheless, significant improvements are needed for the article to be accepted:

We sincerely thank the reviewer for reviewing this manuscript and providing constructive suggestions. We agree that molecular afterglow properties have been extensively reported recently. The suggested reference mainly reported the properties (spectra, photolysis, efficiency etc.) of a series of novel benzylidene iodonium salt one-component photoinitiators that can efficiently decompose under irradiation and their application in 3D printing. However, the main idea of this work is different. Our designed molecules have quite stable and excellent dual-emissive tunable afterglow properties that show a promising application in mechanical self-motoring 3D printing structures, which has not been reported before.

1) The compounds shown are built from a strong donor (DPA) and two acceptors, dibenzothiophene and phosphine oxide. Missing from the text is a reference, which would be a compound built with only DPA and dibenzothiophene, because on the posted results, the contribution of the phosphine group is ambiguous, e.g. the LUMO orbital in both cases is completely localized on dibenzothiophene.

Response: Thanks for the reviewer's invaluable suggestion. Similar properties in compound with a D-A configuration (a compound built with only DPA and dibenzothiophene) have not been reported yet. Actually, we have already performed all the photophysical characterizations of D-A compound (named as DTPA here) previously, which exhibits similar frontier molecular orbital distributions as DTPPAO (Fig. R1a (Fig. S36a)). The redshifted PL spectrum with increasing solvent polarity indicates its singlet state of charge transfer character (Fig. R1b (Fig. S36b)). Here triphenylphosphine (TPP) and diphenyl-sulfone (SF) were chosen as the host matrices. As shown in Fig. R2 (Fig. S37), DTPA/SF doping system shows an obvious RTP emission shoulder in the steady-state emission while RTP emission in DTPA/TPP is relatively weak which is covered by its broad PL emission tail, peaking at 504 nm. The phosphorescence lifetimes of DTPA/TPP and DTPA/SF systems are 4.0 and 149.8 ms, respectively, as shown in Fig. R3. In contrast, both DTPPAO and *t*BuDTPPAO with D-A-A' configuration exhibit obvious RTP emission and longer lifetimes of hundreds of milliseconds in all doping systems, for instance, the lifetime is 110.6 ms in DTPPAO/TPP and 287.7 ms in DTPPAO/SF. It indicates the significance of the additional DPO acceptor unit in DTPPAO and *t*BuDTPPAO.

In addition, the LUMO orbitals of DTPPAO and *t*BuDTPPAO are localized on the weaker DBT (dibenzothiophene) unit, meanwhile LUMO+1 orbitals are mainly localized on the DPO unit (cf. Fig. S54-55). This is mainly due to the energy level degeneracy that LUMO+1 orbital exhibits a similar energy level as LUMO. It gives a potential electron transfer channel from HOMO to LUMO+1, that charge transfer may occur from donor DPA to both acceptor units (DBT and DPO) via either through bond or through space channels, giving stronger RTP properties. Detailed information and figures are shown in Fig. R1-3 in the response to the 2nd comment of the first reviewer.

2) In the methods section, the authors mention that C NMR spectra were recorded for the compounds; unfortunately, the supplement lacks these spectra and their descriptions. They should be added. Moreover, H and C NMR and mass spectra for intermediates I and II should also be added.

Response: We sincerely appreciate the reviewer's suggestion. We are really sorry about the carelessness. The ^1H NMR, ^{13}C NMR and high resolution mass spectra for intermediates **I** and **II** have been added and updated as suggested in Figs. R4-7 and 33-44 (Figs. S1-16).

The following sentences have been added and updated in Page 16 in the Supplementary Information: "The syntheses and structural characterizations of DTPPAO and tBuDTPPAO are presented above in details. Their structures were unambiguously established by a combination of ^1H NMR, ^{13}C NMR spectroscopy, high-resolution mass spectrometry, element analysis (EA) and high performance liquid chromatography (HPLC). The detailed characterizations of their respective intermediates **I** and **II** were also performed. To avoid any possible interference of impurities, strict purification procedures including column chromatography and recrystallization for three times were used."

Fig. R4 (Fig. S3) ^1H NMR spectrum of DTPPAO.

Fig. R5 (Fig. S6) ¹H NMR spectrum of *t*BuDTPPAO.

Fig. R6 (Fig. S9) ¹³C NMR spectrum of DTPPAO.

Fig. R7 (Fig. S12) ¹³C NMR spectrum of *t*BuDTPPAO.

Fig. R33 (Fig. S1) ¹H NMR spectrum of I (DTPPAO).

Fig. R34 (Fig. S2) ¹H NMR spectrum of II (DTPPAO).

Fig. R35 (Fig. S4) ¹H NMR spectrum of I (tBuDTPPAO).

Fig. R36 (Fig. S5) ¹H NMR spectrum of II (tBuDTPPAO).

Fig. R37 (Fig. S7) ¹³C NMR spectrum of I (DTPPAO).

Fig. R38 (Fig. S8) ¹³C NMR spectrum of II (DTPPAO).

Fig. R39 (Fig. S10) ¹³C NMR spectrum of I (tBuDTPPAO).

Fig. R40 (Fig. S11) ¹³C NMR spectrum of **II** (*t*BuDTPPAO).

Fig. R41 (Fig. S13) High resolution mass spectrum of **I** (DTPPAO).

Fig. R42 (Fig. S14) High resolution mass spectrum of II (DTPPAO).

Fig. R43 (Fig. S15) High resolution mass spectrum of I (*t*BuDTPPAO).

Fig. R44 (Fig. S16) High resolution mass spectrum of II (*tBuDTPPAO*).

3) The language needs improvement. The text has numerous phrases, such as "easy processible process" (Page 2, line 51), which should be improved.

Response: Thanks for the reviewer pointing out our mistake in the manuscript. We have carefully checked and made revisions to further improve the manuscript. The changes have been updated and will not influence the content and framework of our work.

Location	Sentences needed correction	Corrected Sentences
Page 2, line 43	due to their unique properties such as structural variety, solution processible fabrication, long emission lifetime and good biocompatibility.	due to their long emission lifetime and high exciton utilization.
Page 2, line 49	In recent decades, enormous efforts have been devoted to stabilize triplet excitons for efficient and long-lived emission,	In recent decades, enormous efforts have been devoted to stabilize triplet excitons,

Page 2, line 52-53	Owing to the versatility of organic molecules and easy processible process	Owing to the structural diversity and processible sample preparation
Page 3, line 81-82	carbon fibers, graphene oxide and reduced graphene oxide, carbon fibers and carbon nanotubes	carbon fibers, graphene oxide and reduced graphene oxide and carbon nanotubes

4) Page 3, lines 79-80 listed twice carbon fibres with two separate references. This should be corrected.

Response: We sincerely thank the reviewer for careful reviewing and suggestion. We are sorry for our carelessness. As suggested by the reviewer, we have carefully checked the manuscript and made the corrections to make it harmonized within the whole manuscript. The careful reviewing indeed improves our manuscript.

The sentence has been changed in Page 3 as follows: "Currently, the most commonly used self-monitoring materials are piezoresistive composites, which are usually achieved by adding conductive materials to polymer matrices, *i.e.*, carbon fibers, graphene oxide and reduced graphene oxide and carbon nanotubes."

5) Add to the supplement graphs the dependence of molar absorption coefficients on wavelength for both compounds.

Response: We thank the reviewer's invaluable comment. Following the reviewer's suggestion, we have added the molar absorption coefficients vs wavelength of compounds DTPPAO and *tBu*DTPPAO as shown in Fig. R45 (Fig. S26). Similar absorption spectra are observed in compounds DTPPAO and *tBu*DTPPAO. Both exhibit two high-energy absorption bands peaking at 246 nm and 305 nm with high extinction coefficients of $4.5 \times 10^5 \text{ M}^{-1} \cdot \text{cm}^{-1}$ and $3 \times 10^5 \text{ M}^{-1} \cdot \text{cm}^{-1}$, respectively, indicating their π - π transition characters. Additionally, they also show a broad low-energy absorption band in the range of 350-400 nm with a smaller extinction coefficient of $\sim 2 \times 10^3 \text{ M}^{-1} \text{ cm}^{-1}$, which can be assigned as n- π character.

Fig. R45 (Fig. S26) Molar extinction coefficient of DTPPAO and *t*BuDTPPAO in 2-methyltetrahydrofuran solution (2-MeTHF, 10^{-5} mol/L) at room temperature.

6) Photolysis diagrams of the presented compounds in solution at the wavelengths of light used in the study should be added to the supplement.

Response: We thank the reviewer's helpful comment. According to the reviewer's suggestion, we have carefully read some papers about photolysis, which is a chemical process in which molecules are split into smaller units through light absorption [18,19]. It is an important tool to identify transient chemical intermediates and to study the mechanisms of fast chemical reactions. In our work, the two designed compounds are quite stable under UV irradiation, undergoing physical processes with electron energized. We reported excellent dual-emissive tunable afterglow properties and demonstrated promising application in mechanical self-motoring 3D printing structures. We think our work doesn't involve the scope of the photolysis, and the study for the photolysis effect may not be necessary. We appreciate your kind suggestion.

References

18. Petko, F., Galek, M., Hola, E., Popielarz, R. and Ortyl, J. One-component cationic photoinitiators from tunable benzylidene scaffolds for 3D printing applications, *Macromolecules* **54**, 7070–7087 (2021).
19. Lee, K., Corrigan, N. and Boyer, C. Rapid high-resolution 3D printing and surface functionalization via type I photoinitiated RAFT polymerization. *Angew. Chem. Int. Ed.* **60**, 8839-8850 (2021).

After reading the manuscript I find it suitable for the journal but the authors are asked to make some clarifications and corrections in the major revision mode.

We sincerely thank again for your time spending on the reviewing and invaluable suggestions to improve our manuscript.

REVIEWERS' COMMENTS

Reviewer #1 (Remarks to the Author):

The authors have resolved my concerns properly. It can be accepted in its current form.

Reviewer #2 (Remarks to the Author):

[Editorial Note: Reviewer #2 was tasked with reviewing and addressing the responses to the concerns raised by Reviewer #3]

The authors have made substantial revisions according to reviewers' comments and all concerns have been addressed. Meanwhile, I also have read the author's answers to the other two reviewers, and the authors have addressed carefully their comments (a large number of experimental data support was added). I recommend its publication in Nature Communications with its current version.